# INCREMENTAL LEARNING IN TRANSFORMERS FOR IN-CONTEXT ASSOCIATIVE RECALL

## ABSTRACT

Transformers acquire in-context learning abilities in abrupt phases during training, often unfolding over multiple stages, during which certain keys circuits like induction heads emerge. In this work, we characterize the training dynamics behind the emergence of such circuits during these stages. We focus on a synthetic in-context associative recall task, where sequences are drawn from random maps between a permutation group and a vocabulary range and the model is required to complete the mapping of a permutation by retrieving it from the context. On this task, we study the trajectories of gradient flow of a simplified two-layer, attention-only transformer. Leveraging symmetries in both the transformer architecture and the data, we derive conservation laws that guide the dynamics of the parameters. These conservation laws crucially reveal how initialization —both in shape and scale— determines the order of learning as well as the timescales over which such circuits emerge revealing the implicit curriculum. Furthermore, at the limit of vanishing scale of initialization, we characterize the trajectory of the gradient flow revealing how the training jumps from one saddle to another. Finally, we provide empirical evidence across different architectural choices, validating our simplifications and generalising the insights from our analysis beyond the simple setting.

## 1 INTRODUCTION

In-context learning (ICL) (Brown, 2020), the ability of a model to perform new tasks from examples provided in its prompt without parameter updates, is a characteristic ability of transformer models. Beyond what these models can do in context, how these abilities emerge during training remains poorly understood. Empirical works Olsson et al. (2022); Chen et al. (2024a) report long plateaus in the training loss followed by abrupt transitions, after which specific circuits, such as induction heads, become functional. Understanding these training dynamics is essential both for theory (identifying which architectural and optimization biases make ICL learnable by gradient descent) and for practice (how hyperparameters convergence speed and training stability).

While recent analyses (Nichani et al., 2024; Chen et al., 2024b) have advanced understanding, they fall short of a complete explanation. Approaches based on layerwise training (Nichani et al., 2024) or highly simplified architectures (Zhang et al., 2025) (e.g., linear attentions) have crucially clarified isolated aspects of the phenomenon, but struggle to account for the sequential acquisition of partial solutions and the duration of plateau phases observed in full models. In particular, we lack a theory that predicts the order in which partial circuits appear, and that explains what controls the length of each phase, including sensitivity to initialization scale and other hyperparameters.

In this paper, we propose combining optimization dynamics with mechanistic interpretability to study how circuits emerge during training. Our analysis is purely dynamical, we study the training trajectories induced by gradient-based optimization, yet our conclusions are mechanistic: we identify which circuit is implemented, which sub-circuits appear first, and how the full circuit crystallizes over time. This interface viewpoint goes beyond static descriptions of trained models by explaining why training lingers on particular plateaus and when it departs them.

To render the problem analytically tractable while preserving its essential structure, we introduce a simplified recall task that retains the induction mechanism underlying in-context n-gram learning but in a form that is more amenable to analysis. Formally, the task is a factual recall: the model

sees a sequence containing a length-$k$ pattern that reappears later, and must match the last $k$ tokens, locate the previous occurrence of that pattern in the context, and predict the next token by recalling the associated response (e.g., contexts of the form $[A][B][C][X] \cdots [A][B][C]$). This task is implemented by the same induction-style circuit as in-context n-grams, but admits cleaner analysis.

Crucially, we couple this task with a series of principled simplifications, each motivated by the circuit that actually solves the task and by typical training behavior. These simplifications isolate the essential components of the transformer responsible for incremental learning, while removing spurious elements that obscure analysis but do not qualitatively affect circuit formation. The resulting model remains rich enough to display the multi-phase, non-convex dynamics observed in practice.

Our analysis reveals a staged learning process: training trajectories encounter intermediate, partially correct solutions where gradients nearly vanish (training plateaus) before transitioning to higher-order solutions. We show that these intermediate solutions align with sub-circuits of the full recall mechanism. This perspective explains both the order in which partial solutions appear and what controls the duration of each phase, providing predictions for the effects of initialization scale and related hyperparameters. Beyond clarifying that induction-like behavior emerges, we characterize how it emerges under gradient-based training.

**Contributions.** Our results (i) formalize a in-context recall task that preserves the induction structure while enabling tractable analysis, (ii) derive training dynamics that exhibit plateaus aligned with sub-circuits, and (iii) provide quantitative predictions for phase ordering and lengths as functions of model and optimization parameters, with empirical validation on small transformers trained end-to-end.

## 1.1 RELATED WORKS

**In-context learning.** The phenomenon of in-context learning (ICL) (Brown, 2020) has been investigated from several perspectives. Mechanistic interpretability has identified induction heads as key circuits supporting ICL (Olsson et al., 2022). A complementary direction examines restricted hypothesis classes, providing controlled settings to analyze how transformers develop in-context capabilities. A recurring observation across these studies is the emergence of training plateaus followed by sudden capability gains (Chen et al., 2024a; Kim et al., 2024). These dynamics have been observed in regression tasks (Garg et al., 2022; Von Oswald et al., 2023; Ahn et al., 2024), boolean and formal language recognition (Bhattamishra et al., 2023; Akyürek et al., 2024), and n-gram prediction.

**$n$-gram models.** $n$-grams models are related our work as the transformer circuit that solves our task also solves the taks of in-context learning $n$-grams. $n$-gram language models (Shannon, 1948; Jurafsky & Martin, 2009) provide a natural testbed for analyzing transformer behavior. Several recent works adopt this viewpoint: the optimization landscape has been analyzed in (Makkuva et al., 2024), expressivity over n-gram distributions in (Svete & Cotterell, 2024), and in-context generalization in (Rajaraman et al., 2024). Other studies connect ICL to the emergence of induction heads Bietti et al. (2024) and their acquisition through gradient descent (Nichani et al., 2024). Edelman et al. (2024) identify stage-wise dynamics in transformer training on in-context n-gram prediction, where intermediate solutions resemble sub-$n$-grams, while Varre et al. (2025) formalize these sub-$n$-grams as near-stationary points. Finally, Chen et al. (2024b) investigate the same task with a modified architecture and initialization scheme that enforces head specialization from the start, thereby eliminating the stage-wise dynamics central to our analysis.

**Incremental learning.** Plateau-shaped learning curves arise broadly in neural network training, beyond ICL. Early work by Fukumizu & Amari (2000) linked such phenomena to critical points in supervised learning. Related characterizations appear in simplified models such as matrix and tensor factorization (Razin et al., 2021; Jiang et al., 2022), matrix sensing (Arora et al., 2019; Li et al., 2021; Jin et al., 2023), diagonal and linear networks (Gissin et al., 2020; Saxe et al., 2019; Gidel et al., 2019; Jacot et al., 2021; Berthier, 2022; Pesme & Flammarion, 2023; Varre et al., 2023; 2024), ReLU networks (Boursier et al., 2022; Abbe et al., 2023), and simplified transformer architectures (Boix-Adsera et al., 2023). Nichani et al. (2025) studied the stage wise dynamics in Factual recall with linear attention. These results provide theoretical tools that we build on to characterize plateaus in in-context learning.

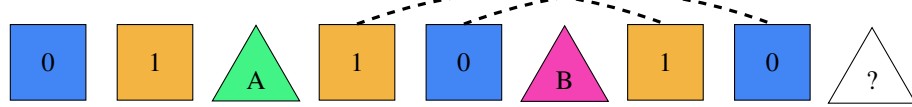

Figure 1: Illustration of the in-context associative recall task. The sequence shows mappings between query permutation elements (rectangles labeled 0, 1) and response tokens (triangles labeled A, B). The model must predict the missing association, marked with "?".

## 2 PROBLEM SETTING

**Notation.** For any positive integers a,b,s, $[s]$ denotes the set $\{0, 1, \ldots, s-1\}$, and $[a, b]$ represents $\{a, \ldots, b\}$. For a vector $v$, its $i$-th coordinate is $v_i$ and $\mathsf{e}_i^s$ is the $i$-th standard basis vector in $\mathbb{R}^s$. For a matrix $A \in \mathbb{R}^{m \times n}$, its entry at row $i$ and column $j$ is $A_{ij}$, and its $r$-th row is $A_r \in \mathbb{R}^n$. For any set $\mathcal{S}$, $|\mathcal{S}|$ denotes the cardinality of the vocabulary. $\Delta^N$ denotes the probability simplex in $\mathbb{R}^N$. The Kronecker product is denoted by $\otimes$. We use $\mathbf{1}$ to denote all vector of all ones.

### 2.1 IN-CONTEXT ASSOCIATIVE RECALL TASK

In-context learning abilities of transformers are driven by certain key circuits, such as induction heads. A basic induction head circuit (Olsson et al., 2022) learns to complete simple repeating patterns, e.g., $[A][B][C] \ldots [A]$. When the model encounters the second $[A]$, the circuit attends to the first $[A]$ and predicts the subsequent token, $[B]$. This paper investigates how models handle a more general version of this pattern: $[A][B][C][X]...[A][B][C]$. In this setting, the model must recognize the entire sequence $[A][B][C]$, locate its previous occurrence in the context, and then use it to predict the next token $[X]$.

Formally, the task is defined as follows. The model must complete a sequence by matching the last $k$ tokens, where $k > 1$ is the *task order*. Let $\mathcal{P}_k$ denote the set of all permutations of $\{0, 1, \ldots, k-1\}$, indexed as $\pi_1, \pi_2, \ldots, \pi_{k!}$, where each $\pi_i$ is a string of $k$ numbers. Let $\mathcal{R}$ be the set of possible responses. The task is defined by a function $f : \mathcal{P}_k \to \mathcal{R}$, sampled from a uniform distribution $\mathcal{D}(\mathcal{F})$ over the set of all such functions $\mathcal{F} = \{f \mid f : \mathcal{P}_k \to \mathcal{R}\}$. An input sequence is then generated by first sampling $q \in \mathcal{P}_k$ uniformly at random, and independently sampling a function $f_\tau$ from $\mathcal{D}(\mathcal{F})$. The final input sequence takes the form:

$$\underbrace{\pi_{(1,0)}, \pi_{(1,1)}, \ldots, \pi_{(1,k-1)}}_{\pi_1}, \; f_\tau(\pi_1), \; \underbrace{\pi_{(2,0)}, \ldots, \pi_{(2,k-1)}}_{\pi_2}, \; f_\tau(\pi_2), \; \ldots, \underbrace{q_0, \ldots, q_{k-1}}_{q_{\mathcal{M}-1}}, \; ?.$$

Note that $\pi_1, \pi_2, \ldots$ each represent a sequence of $k$ tokens, rather than a single token. For example, $\pi_1 = 0, 1, \ldots, k-1$. Figure 1 illustrates the task for $k = 2$ with a response set $\{A, B\}$. We define the vocabulary as $\mathcal{S} = [k] \cup \mathcal{R}$. Each sequence has a fixed length of $l = (k+1)! + k$.

To solve this task, the transformer must learn to identify the part of the context that matches the final $k$ tokens and recall the subsequent token. For completeness, the context contains all possible query permutations, ensuring that the model can always retrieve the correct response, which enables exact learning.

### 2.2 MULTI-HEADED ATTENTION-ONLY TRANSFORMER

We analyze a specific attention-only transformer with a two-layer structure. The first layer contains $k$ attention heads, and the second layer contains a single head. The architecture is based on the disentangled transformer (Friedman et al., 2023; Nichani et al., 2024) and incorporates several simplifications from prior work (Nichani et al., 2024; Edelman et al., 2024; Varre et al., 2025). In particular, tokens are represented with orthogonal encodings, and the only trainable parameters are in the attention layers. We exclude both trainable value matrices and MLP layers. Below, we provide a formal description of this simplified transformer.

**Token encodings.** We represent the input sequence using one-hot encodings. A sequence of length $l$ is mapped into $\mathbb{R}^{|\mathcal{S}|}$ by the embedding function $E : \mathcal{S} \to \mathbb{R}^{|\mathcal{S}|}$, defined as $E(i) = \mathsf{e}_i^{|\mathcal{S}|}$ for $i \in [k]$

and $E(r_i) = \mathbf{e}_{k+i}^{|\mathcal{S}|}$. For convenience, we omit the superscript $|\mathcal{S}|$ in what follows. After the encoding layer, the input sequence $x_0, x_1, \ldots, x_{l-1}$ is given by

$$X = \begin{bmatrix} e_{x_0} & e_{x_1} & \cdots & e_{x_{l-1}} \end{bmatrix}^\top \in \mathbb{R}^{l \times |\mathcal{S}|}.$$

**First attention layer.** The first attention layer has $k$ heads and considers only positional information, which is a convenient choice for this task. We use relative positional encodings with a causal mask. Each head $i$ is parameterized by a vector $\mathbf{w}^i \in \mathbb{R}^l$. The pre-softmax attention scores of head $i$ form a lower-triangular matrix with entries given by

$$\mathbf{A}^i[g, h] = \begin{cases} \mathbf{w}^i_{g-h} & \text{if } 0 \leqslant h \leqslant g \leqslant l-1 \\ -\infty & \text{otherwise} \end{cases}. \tag{1}$$

The output of attention head $i$ ($1 \leqslant i \leqslant k$) is $\mathbf{R}^i = \boldsymbol{\sigma}(\mathbf{A}^i)\, X \in \mathbb{R}^{l \times |\mathcal{S}|}$, where $\boldsymbol{\sigma}$ denotes the row-wise softmax operation with causal masking. The output of the first attention layer is the concatenation of the outputs from all heads together with a skip connection $\mathbf{R}^0 = X$ (which differs from the standard architecture):

$$\mathbf{R} = \begin{bmatrix} \mathbf{R}^0 & \mathbf{R}^1 & \cdots & \mathbf{R}^k \end{bmatrix} \in \mathbb{R}^{l \times (k+1)\mathcal{S}} = \sum_{i=0}^{k} \left(\mathbf{e}_i^{k+1}\right)^\top \otimes \mathbf{R}^i.$$

**Second attention layer.** The second attention layer consists of a single head, parameterized by matrices $\mathbf{Q}, \mathbf{K} \in \mathbb{R}^{(k+1)|\mathcal{S}| \times (k+1)|\mathcal{S}|}$. The attention scores are $\boldsymbol{\sigma}\left(X\mathbf{Q}^\top \mathbf{K} X^\top\right)$, where the softmax is applied row-wise with causal masking. The output of the second layer is

$$\mathbf{R}_+ = \boldsymbol{\sigma}\left(\mathbf{R}\mathbf{Q}^\top \mathbf{K}\mathbf{R}^\top\right)\mathbf{R}V.$$

We choose the value matrix $V = \mathbf{e}_0^{k+1} \otimes \mathrm{I}_{|\mathcal{S}|} \in \mathbb{R}^{(k+1)|\mathcal{S}| \times |\mathcal{S}|}$. This matrix consists of a column of blocks, with the identity matrix as the first block and zeros elsewhere. By construction, $V$ extracts the skip connection from the concatenated output of the first layer, i.e., $\mathbf{R}V = \mathbf{R}^0 = X$, using the mixed-product property of the Kronecker product.

**The model output.** Note that the loss is computed only on the *last token* of the sequence. Consequently, the model's output depends only on the embedding of the final token after the second layer, i.e., $\mathbf{p} = (\mathbf{R}_+)_{l-1} \in \mathbb{R}^{|\mathcal{S}|}$.

$$\begin{aligned} \mathbf{p} = (\mathbf{R}_+)_{l-1} &= \left(\boldsymbol{\sigma}\left(\mathbf{R}\mathbf{Q}^\top \mathbf{K}\mathbf{R}^\top\right) X\right)_{l-1}, \\ &= \mathbf{R}^\top \left(\boldsymbol{\sigma}\left(\mathbf{R}\mathbf{Q}^\top \mathbf{K}\mathbf{R}^\top\right)_{l-1}\right) = \mathbf{R}^\top \boldsymbol{\sigma}\left(\left(\mathbf{R}\mathbf{Q}^\top \mathbf{K}\mathbf{R}^\top\right)_{l-1}\right). \end{aligned}$$

We denote the attention scores by $\mathbf{s} = \boldsymbol{\sigma}\left(\mathbf{R}\mathbf{Q}^\top \mathbf{K}\mathbf{R}^\top\right)_{l-1}$ and the corresponding pre-softmax scores by $\widetilde{\mathbf{s}} = \left(\mathbf{R}\mathbf{Q}^\top \mathbf{K}\mathbf{R}^\top\right)_{l-1}$. The choice of $V$ together with the orthogonal embeddings ensures that $\mathbf{p}$ is a valid probability distribution over the vocabulary, i.e., $\mathbf{p} \in \Delta^{|\mathcal{S}|}$, and requires no further normalization. Hence, the output of the model is given by

$$\mathbf{p} = X^\top \mathbf{s} = X^\top \boldsymbol{\sigma}(\widetilde{\mathbf{s}}).$$

## 2.3 A SIMPLIFIED MODEL

The goal of this paper is to study the training dynamics of transformers on the in-context associative recall task. The simplified architecture described above, although easier than a full transformer, remains too complex for a complete study of training dynamics. To address this intractability, we introduce additional simplifications that preserve the qualitative behavior of the full model while making analysis feasible. Before detailing these simplifications, we first present the construction of the solution implemented by the transformer for this task, which clarifies the rationale behind our design choices.

**The transformer's solution.** To solve the task, the transformer must learn to attend to the portion of the context that matches the final $k$ tokens. This mechanism is implemented through a multi-head construction, variants of which have appeared in prior work (Edelman et al., 2024; Rajaraman et al., 2024). Its parameters are defined as

$$\mathbf{w}^i = c \cdot \mathbf{e}_i^l \text{ where } i \in [1, k], \quad \mathbf{Q}^\top \mathbf{K} = c \left( B \otimes \widetilde{\mathrm{I}}_k \right).$$

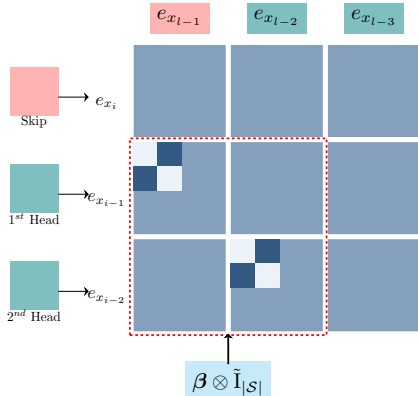

Here $c$ is a positive constant, $\widetilde{\mathrm{I}}_k \in \mathbb{R}^{|\mathcal{S}| \times |\mathcal{S}|}$ where the first $k \times k$ block is given by $\widetilde{\mathrm{I}}_{:k,:k} = \mathrm{I}_k - \gamma \mathbf{1}_k \mathbf{1}_k^\top$ and 0's elsewhere. The matrix $B \in \mathbb{R}^{(k+1) \times (k+1)}$ is defined as

$$\text{For } 0 \leqslant g, h \leqslant k \ B_{gh} = \begin{cases} 1 & \text{if } g+1 = h \\ 0 & \text{otherwise} \end{cases}.$$

As $c \to \infty$, the relative positional encoding ensures that head $h$ outputs the embedding of $x_{i-h}$ for token $i$. With this choice of $B$, the presoftmax attention scores are $\widetilde{\mathbf{s}}_i \approx$

Figure 2: Layer-2 representation structure of the optimal solution constructed by the transfomer We use it to simplify this structure by a diagonal block matrix with trainable scales.

$c \sum_{h=1}^{k} \mathbb{1}\{x_{l-h} = x_{i-h}\}$ which is maximized when the histories of $i$ and $l$ match. In the limit $c \to \infty$, the softmax approaches hardmax attention, i.e, $\mathbf{s}_i \to 1$ (where $i$ is the token that matches the history). The model output is then

$$\mathbf{p} = \sum_j \mathbf{s}_j e_{x_j} \approx e_{x_i}.$$

This construction is illustrated in the Figure 2.

**Simplifying the model.** Our simplifications are motivated by the transformer's solution described above and preserve its overall structure, especially in the second layer. We emphasize that, despite these simplifications, the analysis remains intricate, as shown in the following sections and the training dynamics of the simplified model closely mirror those of the original model. For $i \in [k, l-1], t \in [l], h \in [1, k]$, the modification of the parametric model are given by:

| **Original model** | **Simplified model** |
|---|---|

$$\mathbf{R}_i^h = \sum_{j=0}^{i} \boldsymbol{\sigma}(\mathbf{A}^h)_{ij} e_{x_j}, \qquad \mathbf{R}_i^h = \sum_{j=i-k}^{i-1} \boldsymbol{\sigma}(\mathbf{A}^h)_{ij} e_{x_j},$$

$$\widetilde{\mathbf{s}}_t = \sum_{g,h=0}^{k} \left[ \mathbf{R}_{l-1}^h \right]^\top \left[ \mathbf{Q}^\top \mathbf{K} \right]_{hg} \left[ \mathbf{R}_t^g \right], \qquad \widetilde{\mathbf{s}}_t = \sum_{h=1}^{k} \left[ e_{x_{l-h}} \right]^\top \boldsymbol{\beta}_h^2 \left[ \widetilde{\mathrm{I}}_k \right] \left[ \mathbf{R}_t^h \right],,$$

$$\mathbf{s} = \boldsymbol{\sigma}(\widetilde{\mathbf{s}}), \quad \mathbf{p} = \sum_{t=0}^{l-1} \mathbf{s}_t \, e_{x_t}. \qquad \mathbf{s} = \boldsymbol{\sigma}\left(\widetilde{\mathbf{s}}_\mathcal{R}\right), \quad \mathbf{p} = \sum_{t: x_t \in \mathcal{R}} \mathbf{s}_t \, e_{x_t}.$$

In words, we make the following modifications:

**(A0)** We fix the attention window in the first layer to $k$ for all heads. For each head $h$ we train only the weights $\mathbf{w}_j^h$ for $1 \leqslant j \leqslant k$ while the remaining entries are masked out and set to $-\infty$.

**(A1)** We configure the second-layer attention parameters to match the structure of the optimal solution shown in Figure 2. This configuration is held fixed, and we train only the scalar multipliers that scale these parameters. In particular, we introduce a parameter $\boldsymbol{\beta} \in \mathbb{R}^k$ and parameterize $\mathbf{Q}^\top \mathbf{K}$ as

$$\mathbf{Q}^\top \mathbf{K} = \mathrm{diag}_{-1}\left(\boldsymbol{\beta}^2\right) \otimes \widetilde{\mathrm{I}}_k,$$

where $\mathrm{diag}_{-1}(u) \in \mathbb{R}^{(k+1) \times (k+1)}$ denotes a matrix with $u$ on its first sub-diagonal and zeros elsewhere. Squaring of $\boldsymbol{\beta}$ serves two purposes, (i) it ensures positivity of the entries and (ii) it preserves the 2-homogeneity of $\mathbf{Q}^\top \mathbf{K}$.

(A2) As a further simplification, we replace the embeddings of the last with its embeddings at the solution. This assumption is mild and does not affect the training dynamics. It is mainly a convenience, as it avoids the bilinearity of the first layer outputs and leads to simpler gradient computations.

(A3) Since the output always lies in $\mathcal{R}$, we trim $\widetilde{\mathbf{s}}$ and apply the softmax only to the coordinates corresponding to responses. This ensures the output is always a probability vector over $\mathcal{R}$.

Other than (A1), which simplifies the second attention layer from matrix parameters to vector parameters, the other simplifications are mild and do not affect the essence of the analysis. We justify these choices both analytically and empirically in the next section.

Finally, we denote the parameters of the simplified model by $\boldsymbol{\theta} = (\mathbf{w}^1, \mathbf{w}^2, \ldots, \mathbf{w}^k, \boldsymbol{\beta})$ here $\mathbf{w}^i \in \mathbb{R}^k$ for all $i$ and $\boldsymbol{\beta} = (\boldsymbol{\beta}_1, \boldsymbol{\beta}_2, \ldots, \boldsymbol{\beta}_k) \in \mathbb{R}^k$. We use $\mathbf{p}(\boldsymbol{\theta})$ to denote the output of the model for parameters $\boldsymbol{\theta}$.

### 2.4 THE FINAL PROBLEM SETUP

**Training Objective (A4).** For our analysis, we replace the conventional cross-entropy (CE) loss with the dot-product (DP) loss

$$\ell(\mathbf{p}, \mathbf{p}_*) = 1 - \langle \mathbf{p}, \mathbf{p}_* \rangle \quad \ell_{\mathrm{CE}}(\mathbf{p}, \mathbf{p}_*) = - \langle \mathbf{p}_*, \log \mathbf{p} \rangle.$$

The minimum of the DP loss is

$$\arg\min_{\mathbf{p} \in \Delta^{|\mathcal{S}|}} 1 - \langle \mathbf{p}, \mathbf{p}_* \rangle = e_i \text{ where } i = \arg\max_j (\mathbf{p}_*)_j,$$

while the minimum of the CE loss is $\mathbf{p}_*$. These minima coincide whenever $\mathbf{p}_*$ is a one-hot vector. Since in our task each input sequence has a unique correct response, the target distribution $\mathbf{p}_*$ is always one-hot and the two losses are therefore equivalent. Their gradients also align when $p_*$ is one-hot, differing only by a scaling factor: $\nabla_{\mathbf{p}} \ell(\mathbf{p}, \mathbf{p}_*) = -\mathbf{p}_*$, $\nabla_{\mathbf{p}} \ell_{\mathrm{CE}}(\mathbf{p}, \mathbf{p}_*) = - \langle \mathbf{p}, \mathbf{p}_* \rangle^{-1} \mathbf{p}_*$. Thus, the training dynamics under DP and CE losses are qualitatively identical, making the DP loss a perfect proxy for CE loss in our analysis.

Finally, given the per-sequence DP loss, the population DP loss is

$$\mathcal{L}(\boldsymbol{\theta}) = \mathbb{E}_{f_\tau \sim \mathcal{D}(\mathcal{F}), q \sim \mathcal{P}_k} \ell(\mathbf{p}(\boldsymbol{\theta}), e_{f_\tau(q)}) = 1 - \mathbb{E}_{f_\tau, q} \langle \mathbf{p}(\boldsymbol{\theta}), e_{f_\tau(q)} \rangle. \tag{2}$$

**Gradient Flow.** To analyse the training dynamics, we consider the continuous-time limit of gradient descent, known as gradient flow. The parameters evolve according to the negative gradient of the population loss $\mathcal{L}$ with respect to the parameters. This approach does not account for the stochasticity or adaptive features of the optimizers used in practice. Nevertheless, it captures key aspects of training. The gradient flow is given by

$$\dot{\boldsymbol{\beta}}_h = -\frac{\partial \mathcal{L}}{\partial \boldsymbol{\beta}_h}, \quad \text{and} \quad \dot{\mathbf{w}}_i^h = -\frac{\partial \mathcal{L}}{\partial \mathbf{w}_i^h}, \quad \text{for all } i, h \in [k].$$

## 3 TECHNICAL RESULTS

We study how signals propagate through the layers of a transformer and how the associated training dynamics unfold. Our analysis focuses on the regime of vanishing initialization scale, a setting known for enabling feature learning and favorable generalization properties (Chizat et al., 2019).

**A stagewise learning process.** We train the simplified transformer model with SGD and momentum on the DP loss over the in-context learning task of order $k + 1$. Note that a $k$-head transformer can solve this task exactly: once the model matches the $k$ tokens in a permutation, the remaining token is uniquely determined.

A crucial observation is that the training dynamics are stage-wise. The model plateaus for an extended period before abruptly transitioning to another plateau with lower loss, and eventually converges to zero loss. This behavior becomes more pronounced as the initialization scale decreases, a phenomenon reminiscent of the saddle-to-saddle dynamics observed in deep networks under small-scale initialization (Jacot et al., 2021; Pesme & Flammarion, 2023).

To mechanistically interpret these intermediate stages, we analyze what the model represents on each plateau, see Figure 6. At the first plateau, the model learns to match a single token in the context: one attention head $h_1$ and its associated coefficient $\beta_{h_1}$ are activated. At the second plateau, an additional head $h_2$ with coefficient $\beta_{h_2}$ is activated, enabling the model to match two tokens of the query, $q_{k+1-h_1}, q_{k+1-h_2}$ in the context. This process continues incrementally: at each stage, a new head–coefficient pair is activated, allowing the model to match one additional token. After $k$ such stages, all tokens in the query are matched and the model achieves zero loss.

Formally, if $h_1, h_2, \ldots h_k$ denote the sequence of heads activated across training, then on an input sequence $f_\tau$ the model incrementally learns the functions:

$$f_\tau^\emptyset \longrightarrow f_\tau^{\{h_1\}} \longrightarrow f_\tau^{\{h_1,h_2\}} \longrightarrow \ldots \longrightarrow f_\tau^{[1,k]},$$

where $\emptyset$ is the empty set and $f_\tau^{\mathcal{N}}$ for $\mathcal{N} \subseteq [1,k]$ is a function from $\mathcal{P}_{k+1}$ to $\Delta^{|\mathcal{R}|}$ and gives the frequency of the set $\{f_\tau(\pi) : \forall\, i \in \mathcal{N},\ \pi_{k+1-i} = q_{k+1-i}\}$, i.e., count the frequency of output of the permutations that match $q$ at positions in $\mathcal{N}$ from the right.

**Order of learning.** A key observation concerns the order in which heads are activated. At small initialization scales, this order is determined by the relative magnitudes of the $\beta$ coefficients at initialization. For example, if $\beta_{(h_1)} > \beta_{(h_2)} > \ldots > \beta_{(h_k)}$, the heads are activated sequentially in the order $h_1, h_2, \ldots h_k$, see Fig. 4 in App.. Thus the implicit regularization induced by the scale and shape of initialization provides a natural curriculum, guiding the model to acquire the task in a stage-wise manner. For the remainder of the analysis, we assume without loss of generality that the coefficients are ordered $\beta_1 > \beta_2 > \ldots > \beta_k$, so that heads are activated in order $1, 2, \ldots, k$. By re-indexing the heads, the analysis for arbitrary initial orderings reduces to this canonical case.

We now turn to a detailed study of the stage-wise dynamics. We begin by analyzing a stylized initialization that isolates the transitions between plateaus. We then combine these analyses to obtain a complete picture of the training trajectory.

**The first jump.** We study the dynamics of the first jump, when the model escapes from the initial plateau. We consider a stylized initialization, denoted $\mathcal{I}_1$, where $\beta_1 = \epsilon$ and $\beta_j = 0$ for all $j \neq 1$. The heads are initialized as $\mathbf{w}^i = 0$ for all $i$. Note that at initialization, all heads are symmetric, and there is no signal that breaks this symmetry.

**Theorem 3.1.** *Consider the simplified transformer model $\boldsymbol{\theta} = (\mathbf{w}^1, \mathbf{w}^2, \ldots \mathbf{w}^k, \boldsymbol{\beta})$ with $k$ heads and initialization $\mathcal{I}_1$, evolving under gradient flow on the DP loss. Then:*

(a) **Directional bias:** *For all time $t \geqslant 0$, $\mathbf{w}^1(t) = \alpha_1(t)\mathbf{e}_1^k + \delta_1(t)\mathbf{1}$ for some $\alpha(t), \delta(t) \in \mathbb{R}$.*

(b) **Sparse attention:** $\dot{\mathbf{w}}_1^1 > 0$ *and* $\dot{\mathbf{w}}_i^1 < 0$ *for all $i \neq 1$. That is, the head learns to attend to the first token.*

(c) **A Sufficient ODE:** *The learning dynamics can be fully described by the evolution of $\alpha_1(t), \beta_1(t)$ which satisfy:*

$$\dot{\alpha}_1 = \frac{\beta_1^2 e^{\alpha_1}}{(e^{\alpha_1} + k - 1)^2} \frac{k^2}{k-1} \Xi,$$

$$\dot{\beta}_1 = 2\beta_1 \frac{e^{\alpha_1} - 1}{e^{\alpha_1} + k - 1} \Xi$$

*where*

$$\Xi = \frac{2(1+\gamma)}{(e^{\gamma_1} + k - 1)^2} \left[ \frac{e^{\gamma_1}(|\mathcal{R}| - 1)}{(k-2)!|\mathcal{R}|} \right] \text{ where } \gamma_1 = (1+\gamma)\beta_1^2 \frac{e^{\alpha_1} - 1}{e^{\alpha_1} + k - 1}$$

(d) **Conservation law:** *The quantity $f(\alpha_1) - \beta_1^2/4$ is conserved along the trajectory, i.e., the time derivative $\mathrm{d}(f(\alpha_1) - \beta_1^2/4) = 0$ where*

$$f(\alpha_1) = 2\frac{k-1}{k^2} (\sinh(\alpha_1) - \alpha_1) - \frac{k-1}{k} \left[ e^{-\alpha_1} + \alpha_1 - 1 \right].$$

Some comments are in order.

1. All relative position encodings except $\mathbf{w}_1^1$ (which corresponds to the previous token) evolves together. This follows from the inherent symmetry of the task: since token positions can be permuted within a sequence without leaving the distribution, the dynamics must preserve this symmetry. Combined with the symmetry of the initialization, this leads to the directional bias described above.

2. The transformer rapidly learns to attend to the first token. There is a clear dichotomy: the embedding corresponding to this token grows, while the others decay at proportional rates, leading to rapid sparsification of attention.

3. For this initialization, we obtain an exact ODE description of the dynamics. On any compact set, the time derivatives are bounded away from zero, ensuring that both $\alpha_1$ and $\boldsymbol{\beta}_1$ diverge to infinity, where the gradient is zero. Their evolution is coupled, and a conserved quantity is derived. Conservation laws are common in dynamical systems, and recent work has identified conservation principles in transformers as well (Marcotte et al., 2025). However, prior results are typically restricted to a single attention layer. In contrast, our result shows how parameters across multiple layers, separated by the softmax, jointly obey a conservation law.

4. These conservation laws allow us to derive the timescale of the jump, i.e., how long training remains in the plateau. Suppose $\boldsymbol{\beta}_1(0) = \epsilon$ with $\epsilon \approx 0$. Define $s = e^{\alpha_1} - 1$ with $s(0) = 0$. The conservation law becomes

$$\frac{k-1}{k^2}\left[s+1+\frac{k-1}{s+1}+(k-2)\log(1+s)-k\right]=\frac{\boldsymbol{\beta}_1^2-\epsilon^2}{4}.$$

For small $s$, a Taylor expansion gives $\boldsymbol{\beta}_1^2 \approx 4s^2 + \epsilon^2$, and the local dynamics reduce to

$$\mathrm{d}s \sim c(s^2 + \epsilon^2). \tag{3}$$

Thus the growth of parameters in self-attention has an information exponent of 2 (Arous et al., 2021). Solving the ODE yields $s = \epsilon \tan(\epsilon T)$, implying that $s$ requires $O(1/\epsilon)$ time to reach $O(1)$. Thus, the dynamics spend this amount of time lingering at the plateau.

5. When $s$ has sufficiently grown, the dynamics switch regimes. Now $\boldsymbol{\beta}_h \sim \sqrt{s}$, $\Xi$ which decays in $s$ kicks in and the ODE simplifies to

$$\mathrm{d}s \sim bse^{-s} \implies \int \frac{e^s}{s}\mathrm{d}s = bT \tag{4}$$

The integral function on the right-hand side is the exponential integral, implying that $s$ grows at rate $\log T$ and hence $\alpha_1$ grows only at rate $\log \log T$.

**The subsequent jumps.** Similar to the first jump, we can analyze the subsequent ones. We consider a stylized initialization, denoted $\mathcal{I}_h$, where $\boldsymbol{\beta}_i = c$ for all $i \in [1, h-1]$, $\boldsymbol{\beta}_h = \epsilon$, and $\boldsymbol{\beta}_j = 0$ for all $j > h$, with $c$ taken to be very large. Likewise, we set $\mathbf{w}^i = c_1\mathbf{e}_i^k + c_2\mathbf{1}$ for $i \in [1, h-1]$, and $\mathbf{w}^i = 0$ otherwise. Under this initialization, we study the dynamics of the $h^{\text{th}}$ jump as the model escapes the plateau where it has learned to match $h - 1$ tokens in the context. A key detail is the interplay between macroscopic parameters (the large $c$) and microscopic parameters (the small $\epsilon$). In this setting, the striking feature is that the macroscopic parameters remain stationary while the microscopic ones evolve.

**Proposition 3.2.** *Consider the simplified transformer model $\boldsymbol{\theta} = (\mathbf{w}^1, \mathbf{w}^2, \ldots \mathbf{w}^k, \boldsymbol{\beta})$ with $k$ heads and initialization $\mathcal{I}_h$, evolving under gradient flow on the DP loss. Then:*

(a) ***Stationarity of the macroscopic variables:*** *The gradients of parameters in the first $h - 1$ heads vanish at scale $\nabla_{\mathbf{w}^i}\mathcal{L}, \nabla_{\boldsymbol{\beta}_i}\mathcal{L} = O(e^{-c})$ for $i \in [1, h-1]$.*

(b) ***Dynamics of the microscopic variables:*** *At $c \to \infty$, the dynamics of the remaining parameters corresponds to the those of the first jump on a task of reduced order.*

**Stitching the jumps.** Without loss of generality, assume the initialization $\boldsymbol{\beta} = (c_1\epsilon, c_2\epsilon, c_3\epsilon, \ldots)$, for $c_1 > c_2 > c_3 > \cdots$, with $\epsilon$ very small. Using the first-jump computations, we obtain an time $T$ such that $\boldsymbol{\beta}_1(T) > C$ for some large constant $C$. During this time, the gradients of the other heads remain $O(\epsilon)$, so their parameters stay close to the origin. Next, applying the subsequent-jump

analysis, we can compute a time $T_2$ such that $\boldsymbol{\beta}_2 > C$. Proceeding in this manner, we can stitch the jumps together to describe the full trajectory. This shows that, in principle, the entire evolution can be characterized by chaining together successive jumps, though we do not pursue the full analysis here, as it does not reveal qualitatively new phenomena beyond perturbation analysis.

**Generalizations of the simple model.** We discuss possible relaxations of the perturbed model. In particular, we highlight three illustrative generalizations. For simplification **(A0)**, the attention window of size $k$ provides a convenient way to compute closed-form expressions. The conservation law and the time scale can also be derived without this assumption, though in that case we lose the directional bias and the ability to obtain closed-form formulas. For simplification (A2), replacing the embeddings of the last token does not pose difficulties for the analysis. The argument still holds for the ordering $\boldsymbol{\beta}_1 > \boldsymbol{\beta}_2 > \boldsymbol{\beta}_3 > \cdots$, since the skip connection supplies the embedding of the last token for the first jump, and the head learned at jump $i$ provides the embedding of the last token for jump $i + 1$. Simplification (A3) can also be avoided by choosing a value matrix $V$ that directly outputs the response. However, the output is not guaranteed to be a probability vector. Normalizing by the sum restores this property, making it equivalent to considering the pre-softmax scores of the responses. These generalizations indicate that the phenomena we study are robust to modest relaxations of the simplified setup, even if the algebraic convenience of the original model is lost.

**Experiments.** We use task of order $4$ and the response vocabulary of also size $4$. Overall, the results confirm our theoretical predictions: the model exhibits stage-wise plateaus followed by sharp jumps, with attention heads activating sequentially to implement recall.

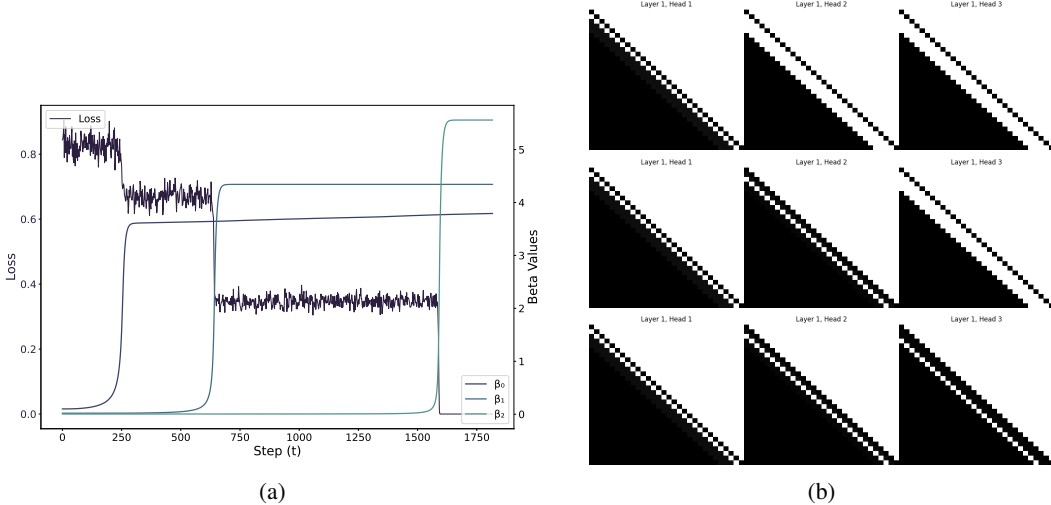

(a)

(b)

Figure 3: The left panel is shown in a and the right panel in b. Note that the $\beta$ values rise simultaneously with the loss, and the attention patterns at times $800$, $1750$, and $4500$ demonstrate incremental learning.

## 4 CONCLUSION

In this work, we analyzed the dynamics of incremental learning in transformers on an in-context associative recall task. By introducing principled simplifications, we constructed a two-layer attention-only architecture where the emergence of recall circuits can be studied exactly. Our main results established conservation laws and stage-wise dynamics that explain the appearance of training plateaus, their dependence on initialization, and the sequential activation of attention heads. These findings offer a mechanistic and dynamical account of how transformers gradually assemble complex recall circuits from simpler sub-circuits. While developed under idealized conditions, extending these results to stochastic optimization, more complex architectures, and finite-sample regimes remains an exciting direction for future work.

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

## A  APPENDIX

**Definition A.1** (Jacobian of a function). Let $f : \mathbb{R}^m \to \mathbb{R}^n$ be a $C_1$-function defined on a variable $X \in \mathbb{R}^m$. $\frac{\partial f}{\partial X}$ denotes the Jacobian which is a function from $\mathbb{R}^m \to \mathbb{R}^{n \times m}$.

## B  PROOFS OF MAIN RESULTS

### B.1  PROOF OF THEOREM 3.1

The gradient flow of the parameters is given by

$$\dot{\boldsymbol{\beta}}_h = -\mathbb{E}\frac{\partial \ell}{\partial \boldsymbol{\beta}_h},$$

$$\dot{\mathbf{w}}_i^h = -\mathbb{E}\frac{\partial \ell}{\partial \mathbf{w}_i^h}.$$

First to show a directional bias, we will show that the trajectory always move along the manifold $\mathcal{M}_1 = \{\boldsymbol{\theta} : \mathbf{w}^1 = \alpha_1 \mathbf{e}_h^1 + \delta_1 \mathbf{1}, \ \alpha_1, \delta_1, \in \mathbb{R} \text{ and } \boldsymbol{\beta}_h, \mathbf{w}^h = 0 \text{ for } h \neq 1\}$.

Consider any point on the manifold $\mathcal{M}_1$ say $\boldsymbol{\theta} = (\mathbf{w}^1, \mathbf{w}^2, \ldots \mathbf{w}^k, \boldsymbol{\beta})$ where $\mathbf{w}^1 = \alpha_1 \mathbf{e}_h^1 + \delta_1 \mathbf{1}$ and $\boldsymbol{\beta}_h = 0$ for $h \neq 1$. We will show that the gradient has no component along the normal space of the manifold $\mathcal{M}_1$ at the point $\boldsymbol{\theta}$. Hence the trajectory will always move along the manifold $\mathcal{M}_1$.

As the parameters satisfy the form prescribed by Lemma C.2, we can invoke the Lemma C.1 to compute the gradients. The population gradients are given by

$$\mathbb{E}\frac{\partial \ell}{\partial \boldsymbol{\beta}_h} = 2\boldsymbol{\beta}_h \frac{e^{\alpha_h} - 1}{e^{\alpha_h} + k - 1}(1 + \gamma)\mathbb{E}\Delta_{h,h}, \tag{5}$$

$$\mathbb{E}\frac{\partial \ell}{\partial \mathbf{w}_i^h} = (1 + \gamma)\frac{\boldsymbol{\beta}_h^2 \left((e^{\alpha_h} - 1)\,\mathbb{1}\{i = h\} + 1\right)}{e^{\alpha_h} + k - 1}\left(\mathbb{E}\Delta_{h,i} - \frac{e^{\alpha_h} - 1}{e^{\alpha_h} + k - 1}\mathbb{E}\Delta_{h,h}\right) \tag{6}$$

where

$$\Delta_{h,i} = - \left[ \sum_{t=1}^{k!} \mathbf{s}_t \mathbb{1}\{r_t = r \wedge q_{k-h} = \pi_{(t,k-i)}\} - \left( \sum_{j=1}^{k!} \mathbf{s}_j \mathbb{1}\{r_j = r\} \right) \left( \sum_{t=1}^{k!} \mathbf{s}_t \left( \mathbb{1}\{q_{k-h} = \pi_{(t,k-i)}\} \right) \right) \right],$$

$$s = \boldsymbol{\sigma}(\mathbf{s}^{'}), \quad \mathbf{s}_t^{'} = \sum_{h=1}^{k} \gamma_h \mathbb{1}\{q_{k-h} = \pi_{(t,k-h)}\}$$

$$\gamma_h = (1 + \gamma)\boldsymbol{\beta}_h^2 \frac{e^{\alpha_h} - 1}{e^{\alpha_h} + k - 1}$$

Note that the gradients of $\boldsymbol{\beta}_h$ for $h \neq 1$ are zero as $\boldsymbol{\beta}_h = 0$ for $h \neq 1$. Also note that the gradients of $\mathbf{w}^h$ for $h \neq 1$ are zero as $\boldsymbol{\beta}_h^2 = 0$ for $h \neq 1$. The only thing that is left to show is that the gradients of $\mathbf{w}^1$ are of the form $\frac{\partial \ell}{\partial \mathbf{w}^1} = \xi_1 \mathbf{e}_1^1 + \zeta_1 \mathbf{1}$ for some $\xi_1, \zeta_1 \in \mathbb{R}$.

Note that $\mathbf{s}_t^{'} = \gamma_1 \mathbb{1}\{q_{k-1} = \pi_{(t,k-1)}\}$ as $\gamma_h = 0$ for $h \neq 1$ and the softmax score is given by

$$\mathbf{s}_t = \frac{e^{\gamma_1 \mathbb{1}\{q_{k-1}=\pi_{(t,k-1)}\}}}{\sum_{j=1}^{k!} e^{\gamma_1 \mathbb{1}\{q_{k-1}=\pi_{(j,k-1)}\}}} = \frac{(e^{\gamma_1} - 1)\mathbb{1}\{q_{k-1} = \pi_{(t,k-1)}\} + 1}{e^{\gamma_1} + k - 1} \frac{1}{(k-1)!}.$$

From Lemma C.3 we have

$$\mathbb{E}\Delta_{1,i} = \frac{1}{(e^{\gamma_1} + k - 1)^2} \left[ \frac{e^{\gamma_1}(|\mathcal{R}| - 1)}{(k-1)!|\mathcal{R}|} \right], \quad \text{for } i \neq 1.$$

$$\mathbb{E}\Delta_{1,1} = -(k-1)\mathbb{E}\Delta_{1,i} = -\frac{1}{(e^{\gamma_1} + k - 1)^2} \left[ \frac{e^{\gamma_1}(|\mathcal{R}| - 1)}{(k-2)!|\mathcal{R}|} \right].$$

So the gradient of $\mathbf{w}_i^1$ given by $\frac{\partial \ell}{\partial \mathbf{w}_i^1}$ is independent of $i$ for $i \neq 1$ since $\mathbb{E}\Delta_{1,i}$ is independent of $i$. Hence the gradient of $\mathbf{w}^1$ is of the form $\frac{\partial \ell}{\partial \mathbf{w}^1} = \xi_1 \mathbf{e}_1^1 + \zeta_1 \mathbf{1}$ for some $\xi_1, \zeta_1 \in \mathbb{R}$. This proves the directional bias of the trajectory.

For the next signal propagation part of the result, we will show that there exists note that $\mathbb{E}\Delta_{1,1} < 0$ for all $\gamma_1 > 0$ and $\mathbb{E}\Delta_{1,i} > 0$ for all $\gamma_1 > 0$ and $i \neq 1$. Using these expressions, the population gradient of $\mathbf{w}_1^1$ is given by

$$\mathbb{E}\frac{\partial \ell}{\partial \mathbf{w}_1^1} = (1 + \gamma)\frac{\boldsymbol{\beta}_1^2 e^{\alpha_1}}{e^{\alpha_1} + k - 1} \left( \mathbb{E}\Delta_{1,1} - \frac{e^{\alpha_1} - 1}{e^{\alpha_1} + k - 1}\mathbb{E}\Delta_{1,1} \right)$$

$$= (1 + \gamma)\frac{\boldsymbol{\beta}_1^2 e^{\alpha_1}}{e^{\alpha_1} + k - 1} \left( \frac{k - 1}{e^{\alpha_1} + k - 1}\mathbb{E}\Delta_{1,1} \right) < 0$$

Similarly the population gradient of $\mathbf{w}_i^1$ for $i \neq 1$ is given by

$$\mathbb{E}\frac{\partial \ell}{\partial \mathbf{w}_i^1} = (1 + \gamma)\frac{\boldsymbol{\beta}_1^2}{e^{\alpha_1} + k - 1} \left( \mathbb{E}\Delta_{1,i} - \frac{e^{\alpha_1} - 1}{e^{\alpha_1} + k - 1}\mathbb{E}\Delta_{1,1} \right),$$

$$= (1 + \gamma)\frac{\boldsymbol{\beta}_1^2}{e^{\alpha_1} + k - 1} \left( -\frac{\mathbb{E}\Delta_{1,1}}{k - 1} - \frac{e^{\alpha_1} - 1}{e^{\alpha_1} + k - 1}\mathbb{E}\Delta_{1,1} \right),$$

$$> 0$$

The dynamics of gradient descent is given by

$$\dot{\mathbf{w}}_1^1 = -\mathbb{E}\frac{\partial \ell}{\partial \mathbf{w}_1^1} = -(1 + \gamma)\frac{\boldsymbol{\beta}_1^2 e^{\alpha_1}}{e^{\alpha_1} + k - 1} \left( \frac{k}{e^{\alpha_1} + k - 1}\mathbb{E}\Delta_{1,1} \right) > 0,$$

$$\dot{\mathbf{w}}_i^1 = -\mathbb{E}\frac{\partial \ell}{\partial \mathbf{w}_i^1} = (1 + \gamma)\frac{\boldsymbol{\beta}_1^2}{e^{\alpha_1} + k - 1} \left( -\frac{\mathbb{E}\Delta_{1,1}}{k - 1} - \frac{e^{\alpha_1} - 1}{e^{\alpha_1} + k - 1}\mathbb{E}\Delta_{1,1} \right) < 0, \quad i \neq 1$$

$$\dot{\boldsymbol{\beta}}_1 = -\mathbb{E}\frac{\partial \ell}{\partial \boldsymbol{\beta}_1} = -2\boldsymbol{\beta}_1 \frac{e^{\alpha_1} - 1}{e^{\alpha_1} + k - 1}(1 + \gamma)\mathbb{E}\Delta_{1,1} > 0$$

The complete dynamics of the system is given by $\dot{\alpha}_1, \dot{\beta}_1$ and they can be written as

$$\dot{\alpha}_1 = \dot{\mathbf{w}}_1^1 - \dot{\mathbf{w}}_2^1 = (1+\gamma)\frac{\boldsymbol{\beta}_1^2}{e^{\alpha_1}+k-1}\left(\frac{(k)e^{\alpha_1}}{e^{\alpha_1}+k-1}\mathbb{E}\Delta_{1,1} + \frac{\mathbb{E}\Delta_{1,1}}{k-1} + \frac{e^{\alpha_1}-1}{e^{\alpha_1}+k-1}\mathbb{E}\Delta_{1,1}\right) < 0,$$

$$\dot{\boldsymbol{\beta}}_1 = -2\boldsymbol{\beta}_1\frac{e^{\alpha_1}-1}{e^{\alpha_1}+k-1}(1+\gamma)\mathbb{E}\Delta_{1,1} > 0$$

The final system of ODE is given by '

$$\dot{\alpha}_1 = \frac{\boldsymbol{\beta}_1^2 e^{\alpha_1}}{e^{\alpha_1}+k-1}^2 \frac{k^2}{k-1} - \mathbb{E}(1+\gamma)\Delta_{1,1},$$

$$\dot{\boldsymbol{\beta}}_1 = 2\boldsymbol{\beta}_1\frac{e^{\alpha_1}-1}{e^{\alpha_1}+k-1} - \mathbb{E}(1+\gamma)\Delta_{1,1}$$

Lets denote $\Xi = -2(1+\gamma)\mathbb{E}\Delta_{1,1}$ which is positive and given by

$$\Xi = \frac{2(1+\gamma)}{(e^{\gamma_1}+k-1)^2}\left[\frac{e^{\gamma_1}(|\mathcal{R}|-1)}{(k-2)!|\mathcal{R}|}\right] \text{ where } \gamma_1 = (1+\gamma)\boldsymbol{\beta}_1^2\frac{e^{\alpha_1}-1}{e^{\alpha_1}+k-1}$$

Using this the system of ODE can be written as

$$\dot{\alpha}_1 = \frac{\boldsymbol{\beta}_1^2 e^{\alpha_1}}{(e^{\alpha_1}+k-1)^2}\frac{k^2}{k-1}\Xi,$$

$$\dot{\boldsymbol{\beta}}_1 = 2\boldsymbol{\beta}_1\frac{e^{\alpha_1}-1}{e^{\alpha_1}+k-1}\Xi$$

Note that

$$\frac{\dot{\boldsymbol{\beta}}_1^2}{4} = \boldsymbol{\beta}_1^2\frac{e^{\alpha_1}-1}{e^{\alpha_1}+k-1}\Xi$$

$$\dot{f(\alpha_1)} = f'(\alpha_1)\dot{\alpha}_1 = f'(\alpha_1)\frac{\boldsymbol{\beta}_1^2 e^{\alpha_1}}{(e^{\alpha_1}+k-1)^2}\frac{k^2}{k-1}\Xi$$

If we choose $f(\alpha_1)$ such that

$$f'(\alpha_1)\frac{\boldsymbol{\beta}_1^2 e^{\alpha_1}}{(e^{\alpha_1}+k-1)^2}\frac{k^2}{k-1}\Xi = \boldsymbol{\beta}_1^2\frac{e^{\alpha_1}-1}{e^{\alpha_1}+k-1}\Xi,$$

$$\implies f'(\alpha_1) = \frac{k-1}{k^2}\frac{(e^{\alpha_1}-1)(e^{\alpha_1}-1+k)}{e^{\alpha_1}}$$

$$= \frac{k-1}{k^2}\left(e^{\alpha_1} - 2 + e^{-\alpha_1} - ke^{-\alpha_1} + k\right)$$

$$\implies f(\alpha_1) = \frac{k-1}{k^2}\left(e^{\alpha_1} - 2\alpha_1 - e^{-\alpha_1} + ke^{-\alpha_1} + k\alpha_1\right),$$

$$f(\alpha_1) = 2\frac{k-1}{k^2}\left(\sinh(\alpha_1) - \alpha_1\right) + \frac{k-1}{k}\left[e^{-\alpha_1} + \alpha_1 - 1\right]$$

Now $f(\alpha_1) - \boldsymbol{\beta}_1^2/4$ is conserved along the trajectory. As $\mathrm{d}(f(\alpha_1) - \boldsymbol{\beta}_1^2/4) = 0$.

This completes the proof of the theorem.

## B.2 PROOF OF THEOREM 3.2

The gradient flow is given by The population gradients are given by

$$\mathbb{E}\frac{\partial\ell}{\partial\boldsymbol{\beta}_h} = 2\boldsymbol{\beta}_h\frac{e^{\alpha_h}-1}{e^{\alpha_h}+k-1}(1+\gamma)\mathbb{E}\Delta_{h,h}, \tag{7}$$

$$\mathbb{E}\frac{\partial\ell}{\partial\mathbf{w}_i^h} = (1+\gamma)\frac{\boldsymbol{\beta}_h^2\left((e^{\alpha_h}-1)\mathbb{1}\{i=h\}+1\right)}{e^{\alpha_h}+k-1}\left(\mathbb{E}\Delta_{h,i} - \frac{e^{\alpha_h}-1}{e^{\alpha_h}+k-1}\mathbb{E}\Delta_{h,h}\right) \tag{8}$$

where

$$\Delta_{h,i} = -\left[\sum_{t=1}^{k!}\mathbf{s}_t\mathbb{1}\{r_t = r \wedge q_{k-h} = \pi_{(t,k-i)}\} - \left(\sum_{j=1}^{k!}\mathbf{s}_j\mathbb{1}\{r_j = r\}\right)\left(\sum_{t=1}^{k!}\mathbf{s}_t\left(\mathbb{1}\{q_{k-h} = \pi_{(t,k-i)}\}\right)\right)\right],$$

$$s = \boldsymbol{\sigma}(\mathbf{s}'), \quad \mathbf{s}'_t = \sum_{h=1}^{k}\gamma_h\mathbb{1}\{q_{k-h} = \pi_{(t,k-h)}\}$$

$$\gamma_h = (1+\gamma)\boldsymbol{\beta}_h^2\frac{e^{\alpha_h} - 1}{e^{\alpha_h} + k - 1}$$

Without loss of generality, assume that $\gamma_i = c$ for $i \in [1, h-1]$. Now split the set of permutations into two partitions $\mathcal{P}_S$ and $\mathcal{P}_S^c$ where for query $q$, $\mathcal{P}_S$ is the set that matches the last $h-1$ tokens and others do not. Now $\mathbf{s}_t = e^{-c}$ for $t \in \mathcal{P}_S^c$. Now $\Delta_{i,j}$ for $i \in [1, h-1]$ and $j \neq i$, it can seen that $\mathbf{s}_t\left(\mathbb{1}\{q_{k-i} = \pi_{(t,k-j)}\}\right) = e^{-c}$ for any $t$, hence $\mathbb{E}\Delta_{i,j} = e^{-c}$. The cases left are $\mathbb{E}\Delta_{i,i}$, but it is also of $e^{-c}$ as the $\Delta_{i,:}$'s sum to 0.

Now the case of $i = h$, now for $t \in \mathcal{P}_S$

$$s_t = \frac{\exp\{c\}\exp\{\gamma_h\mathbb{1}\{q_{k-h} = \pi_{(t,k-h)}\}\}}{\sum_{\pi_t \in \mathcal{P}_S}\exp\{c\}\exp\{\gamma_h\mathbb{1}\{q_{k-h} = \pi_{(t,k-h)}\}\} + O(1)},$$

$$= \frac{\exp\{\gamma_h\mathbb{1}\{q_{k-h} = \pi_{(t,k-h)}\}\}}{\sum_{\pi_t \in \mathcal{P}_S}\exp\{\gamma_h\mathbb{1}\{q_{k-h} = \pi_{(t,k-h)}\}\}} + O(e^{-c})$$

This expression is exactly equivalent to the first jump however now restricted on the set $P_S$ of reduced order.

## C COMPUTATIONS OF THE DERIVATIVES OF THE SIMPLIFIED TRANSFORMER MODEL

The forward pass of the simplified transformer model on a single input sequence is given by, let $r$ be the response of the query.

$$\mathbf{R}_i^h = \sum_{j=i-k}^{i-1}\boldsymbol{\sigma}(\mathbf{A}^h)_{ij}\,e_{x_j}, \tag{9a}$$

$$\widetilde{\mathbf{s}}_t = \sum_{h=1}^{k}\boldsymbol{\beta}_h^2\left\langle e_{x_{l-h}}, \widetilde{\mathbf{I}}_k\,\mathbf{R}_t^h\right\rangle, \tag{9b}$$

$$\mathbf{s} = \boldsymbol{\sigma}\left(\widetilde{\mathbf{s}}_\mathcal{R}\right) \tag{9c}$$

$$\mathbf{p} = \sum_{t:x_t \in \mathcal{R}}\mathbf{s}_t\,e_{x_t} \tag{9d}$$

$$\ell = 1 - \langle\mathbf{p}, e_r\rangle \tag{9e}$$

For the convenience of the analysis, we will drop $\mathcal{R}$ in the subscript of $\widetilde{\mathbf{s}}$ and now $\widetilde{\mathbf{s}} \in \mathbb{R}^{k!}$ where $\widetilde{\mathbf{s}}_t$ denotes the score of the response corresponding to the $t^{th}$ permutation in the sequence.

**Derivatives wrt to the $\widetilde{\mathbf{s}}$.** The derivatives of the loss with respect to the predicted scores can be computed using the chain rule:

$$\frac{\partial\ell}{\partial\widetilde{\mathbf{s}}} = \frac{\partial\ell}{\partial\mathbf{p}}\frac{\partial\mathbf{p}}{\partial\mathbf{s}}\frac{\partial\mathbf{s}}{\partial\widetilde{\mathbf{s}}} = -e_r^\top X_\mathcal{R}^\top\left(\mathrm{diag}(\mathbf{s}) - \mathbf{s}\,\mathbf{s}^\top\right).$$

The co-ordinate wise derivative is

$$\frac{\partial\ell}{\partial\widetilde{\mathbf{s}}_t} = -\mathbf{s}_t\left(\mathbb{1}\{r_t = r\} - \sum_{j=1}^{k!}\mathbf{s}_j\mathbb{1}\{r_j = r\}\right) = \ell_t'.$$

where $r_t$ is the response corresponding to the $t^{th}$ permutation in the sequence. Using the property of the softmax we have $\ell(\widetilde{\mathbf{s}} + c\mathbf{1}) = \ell(\widetilde{\mathbf{s}})$ as the softmax is invariant when even co-ordinate is shifted by a constant. Now taking the derivative with $c$ at $c = 0$ we get

$$\sum_{t=1}^{k!} \ell'_t = 0 \tag{10}$$

**Derivative of $\widetilde{\mathbf{s}}$ wrt to $\boldsymbol{\beta}$ and $\mathbf{w}^h$'s** The derivative of $\widetilde{\mathbf{s}}$ with respect to $\boldsymbol{\beta}$ can be computed as follows:

$$\frac{\partial \widetilde{\mathbf{s}}_t}{\partial \boldsymbol{\beta}_h} = 2\boldsymbol{\beta}_h \left\langle e_{x_{l-h}}, \widetilde{\mathbf{I}}_k \mathbf{R}_t^h \right\rangle$$

The derivative of $\widetilde{\mathbf{s}}$ with respect to $\mathbf{w}_i^h$ can be computed as follows:

$$\frac{\partial \widetilde{\mathbf{s}}_t}{\partial \mathbf{w}_i^h} = \boldsymbol{\beta}_h^2 \left\langle e_{x_{l-h}}, \widetilde{\mathbf{I}}_k \frac{\partial \mathbf{R}_t^h}{\partial \mathbf{w}_i^h} \right\rangle$$

Using Lemma D.2 we have

$$\frac{\partial \widetilde{\mathbf{s}}_t}{\partial \mathbf{w}_i^h} = \boldsymbol{\beta}_h^2 \left\langle e_{x_{l-h}}, \widetilde{\mathbf{I}}_k \frac{e^{\mathbf{w}_i^h}}{\sum_{j=1}^k e^{\mathbf{w}_j^h}} \left(e_{\pi_{(t,k-i)}} - \mathbf{R}_t^h\right) \right\rangle$$

$$= \boldsymbol{\beta}_h^2 \frac{e^{\mathbf{w}_i^h}}{\sum_{j=1}^k e^{\mathbf{w}_j^h}} \left( \left\langle e_{x_{l-h}}, \widetilde{\mathbf{I}}_k \, e_{\pi_{(t,k-i)}} \right\rangle - \left\langle e_{x_{l-h}}, \widetilde{\mathbf{I}}_k \mathbf{R}_t^h \right\rangle \right)$$

Note that $x_{l-h} = q_{k-h}$ and $\left\langle e_{x_{l-h}}, \widetilde{\mathbf{I}}_k \, e_{\pi_{(t,k-i)}} \right\rangle = \mathbb{1}\{q_{k-h} = \pi_{(t,k-i)}\} - \gamma\mathbb{1}\{q_{k-h} \neq \pi_{(t,k-i)}\}\mathbb{1}\{q_{k-h} = \pi_{(t,k-i)}\} - \gamma$. Using this computation, we have,

$$\frac{\partial \widetilde{\mathbf{s}}_t}{\partial \mathbf{w}_i^h} = \frac{\boldsymbol{\beta}_h^2 e^{\mathbf{w}_i^h}}{\sum_{j=1}^k e^{\mathbf{w}_j^h}} \left( (1+\gamma)\mathbb{1}\{q_{k-h} = \pi_{(t,k-i)}\} - \gamma - \left\langle e_{x_{l-h}}, \widetilde{\mathbf{I}}_k \mathbf{R}_t^h \right\rangle \right)$$

Computing the derivative of the loss with respect to $\boldsymbol{\beta}$ and $\mathbf{w}_i^h$ we have

$$\frac{\partial \ell}{\partial \boldsymbol{\beta}_h} = \sum_{t=1}^{k!} \ell'_t \, 2\boldsymbol{\beta}_h \left\langle e_{x_{l-h}}, \widetilde{\mathbf{I}}_k \mathbf{R}_t^h \right\rangle = 2\boldsymbol{\beta}_h \sum_{t=1}^{k!} \ell'_t \left\langle e_{x_{l-h}}, \widetilde{\mathbf{I}}_k \mathbf{R}_t^h \right\rangle$$

$$\frac{\partial \ell}{\partial \mathbf{w}_i^h} = \sum_{t=1}^{k!} \ell'_t \, \frac{\boldsymbol{\beta}_h^2 e^{\mathbf{w}_i^h}}{\sum_{j=1}^k e^{\mathbf{w}_j^h}} \left( (1+\gamma)\mathbb{1}\{q_{k-h} = \pi_{(t,k-i)}\} - \gamma - \left\langle e_{x_{l-h}}, \widetilde{\mathbf{I}}_k \mathbf{R}_t^h \right\rangle \right),$$

$$= \frac{\boldsymbol{\beta}_h^2 e^{\mathbf{w}_i^h}}{\sum_{j=1}^k e^{\mathbf{w}_j^h}} \sum_{t=1}^{k!} \ell'_t \left( (1+\gamma)\mathbb{1}\{q_{k-h} = \pi_{(t,k-i)}\} - \left\langle e_{x_{l-h}}, \widetilde{\mathbf{I}}_k \mathbf{R}_t^h \right\rangle \right),$$

Now collecting the final expressions we have the following lemma.

**Lemma C.1.** *The derivatives of the loss $\ell$ with respect to the parameters $\boldsymbol{\beta}$ and $\mathbf{w}^h$'s are given by*

$$\frac{\partial \ell}{\partial \boldsymbol{\beta}_h} = 2\boldsymbol{\beta}_h \sum_{t=1}^{k!} \ell'_t \left\langle e_{x_{l-h}}, \widetilde{\mathbf{I}}_k \mathbf{R}_t^h \right\rangle$$

$$\frac{\partial \ell}{\partial \mathbf{w}_i^h} = \frac{\boldsymbol{\beta}_h^2 e^{\mathbf{w}_i^h}}{\sum_{j=1}^k e^{\mathbf{w}_j^h}} \sum_{t=1}^{k!} \ell'_t \left( (1+\gamma)\mathbb{1}\{q_{k-h} = \pi_{(t,k-i)}\} - \left\langle e_{x_{l-h}}, \widetilde{\mathbf{I}}_k \mathbf{R}_t^h \right\rangle \right),$$

*where $\ell'_t = -\mathbf{s}_t \left( \mathbb{1}\{r_t = r\} - \sum_{j=1}^{k!} \mathbf{s}_j \mathbb{1}\{r_j = r\} \right)$.*

The proof of the lemma is given above.

Now we will use the above lemma to compute the gradients at a particular parameter configuration, we choose a general parameter configuration so that we can invoke this lemma whenever gradient computations are needed.

**Lemma C.2.** *Consider a parameter configuration $\mathcal{I}_{\mathcal{G}}$ defined such that for all $h$ and $i$,*

$$\mathbf{w}^h = \alpha_h \mathbf{e}_h^k + \delta_h \mathbf{1}$$

*$\boldsymbol{\beta}_h$ is used as is. Then the gradients at this parameter configuration are given by*

$$\frac{\partial \ell}{\partial \boldsymbol{\beta}_h} = 2\boldsymbol{\beta}_h \frac{e^{\alpha_h} - 1}{e^{\alpha_h} + k - 1}(1+\gamma)\Delta_{h,h}, \tag{11}$$

$$\frac{\partial \ell}{\partial \mathbf{w}_i^h} = (1+\gamma)\frac{\boldsymbol{\beta}_h^2 e^{\alpha_h + \delta_h}}{e^{\alpha_h} + k - 1}\left(\Delta_{h,i} - \frac{e^{\alpha_h} - 1}{e^{\alpha_h} + k - 1}\Delta_{h,h}\right) \tag{12}$$

*where*

$$\Delta_{h,i} = -\left[\sum_{t=1}^{k!} \mathbf{s}_t \mathbb{1}\{r_t = r \wedge q_{k-h} = \pi_{(t,k-i)}\} - \left(\sum_{j=1}^{k!} \mathbf{s}_j \mathbb{1}\{r_j = r\}\right)\left(\sum_{t=1}^{k!} \mathbf{s}_t \left(\mathbb{1}\{q_{k-h} = \pi_{(t,k-i)}\}\right)\right)\right]$$

*where $s = \boldsymbol{\sigma}(\mathbf{s}')$ and $\mathbf{s}_t' = \sum_{h=1}^{k} \gamma_h \mathbb{1}\{q_{k-h} = \pi_{(t,k-h)}\}$ with $\gamma_h = (1+\gamma)\boldsymbol{\beta}_h^2 \frac{e^{\alpha_h}-1}{e^{\alpha_h}+k-1}$.*

*Proof.* First let us compute the forward pass

$$\mathbf{R}_t^h = \sum_{i=1}^{k} \boldsymbol{\sigma}(\mathbf{A}^h)_{t,t-i}\, e_{\pi_{(t,k-i)}} = \sum_{i=1}^{k} \frac{e^{\mathbf{w}_i^h}}{\sum_{j=1}^{k} e^{\mathbf{w}_j^h}}\, e_{\pi_{(t,k-i)}},$$

$$= \frac{e^{\alpha_h + \delta_h}}{(e^{\alpha_h} + k - 1)\,e^{\delta_h}}\, e_{\pi_{(t,k-h)}} + \sum_{i \neq h} \frac{e^{\delta_h}}{(e^{\alpha_h} + k - 1)\,e^{\delta_h}}\, e_{\pi_{(t,k-i)}},$$

$$= \frac{e^{\alpha_h} - 1}{e^{\alpha_h} + k - 1}e_{\pi_{(t,k-h)}} + \frac{1}{e^{\alpha_h} + k - 1}\mu$$

Note that $\mu = \sum_{i=1}^{k} e_{\pi_{(t,k-i)}}$ is the same for all $t$. Now the presoftmax score is given by

$$\widetilde{\mathbf{s}}_t = \sum_{h=1}^{k} \boldsymbol{\beta}_h^2 \left\langle e_{x_{l-h}}, \widetilde{\mathbf{I}}_k\, \mathbf{R}_t^h \right\rangle,$$

$$= \sum_{h=1}^{k} \boldsymbol{\beta}_h^2 \left(\frac{e^{\alpha_h} - 1}{e^{\alpha_h} + k - 1}(1+\gamma)\mathbb{1}\{q_{k-h} = \pi_{(t,k-h)}\} - \gamma - \frac{1}{e^{\alpha_h} + k - 1}(1+\gamma)\right),$$

$$= (1+\gamma)\sum_{h=1}^{k} \boldsymbol{\beta}_h^2 \frac{e^{\alpha_h} - 1}{e^{\alpha_h} + k - 1}\mathbb{1}\{q_{k-h} = \pi_{(t,k-h)}\} - (1+\gamma)\sum_{h=1}^{k} \frac{\boldsymbol{\beta}_h^2}{e^{\alpha_h} + k - 1} - \gamma \sum_{h=1}^{k} \boldsymbol{\beta}_h^2.$$

Define

$$\gamma_h = (1+\gamma)\boldsymbol{\beta}_h^2 \frac{e^{\alpha_h} - 1}{e^{\alpha_h} + k - 1}. \tag{13}$$

Denote

$$\mathbf{s}_t' = \sum_{h=1}^{k} \gamma_h \mathbb{1}\{q_{k-h} = \pi_{(t,k-h)}\}.$$

Note that $\widetilde{\mathbf{s}}_t = \mathbf{s}'_t + c$ where $c$ is a constant independent of $t$. Using the property of the softmax we have $\mathbf{s} = \ell(\mathbf{s}')$. Using the lemma C.1 we have

$$\frac{\partial \ell}{\partial \boldsymbol{\beta}_h} = 2\boldsymbol{\beta}_h \sum_{t=1}^{k!} \ell'_t \left\langle e_{x_{l-h}}, \widetilde{\mathrm{I}}_k \, \mathbf{R}^h_t \right\rangle,$$

$$= 2\boldsymbol{\beta}_h \sum_{t=1}^{k!} \ell'_t \left( \frac{e^{\alpha_h} - 1}{e^{\alpha_h} + k - 1}(1+\gamma)\mathbb{1}\{q_{k-h} = \pi_{(t,k-h)}\} - \gamma - \frac{1}{e^{\alpha_h} + k - 1}(1+\gamma) \right),$$

$$= 2\boldsymbol{\beta}_h \frac{e^{\alpha_h} - 1}{e^{\alpha_h} + k - 1}(1+\gamma) \sum_{t=1}^{k!} \ell'_t \mathbb{1}\{q_{k-h} = \pi_{(t,k-h)}\} - 2\boldsymbol{\beta}_h \left( \gamma + \frac{1}{e^{\alpha_h} + k - 1}(1+\gamma) \right) \sum_{t=1}^{k!} \ell'_t,$$

$$= 2\boldsymbol{\beta}_h \frac{e^{\alpha_h} - 1}{e^{\alpha_h} + k - 1}(1+\gamma) \sum_{t=1}^{k!} \ell'_t \mathbb{1}\{q_{k-h} = \pi_{(t,k-h)}\},$$

Now the derivative with respect to $\mathbf{w}^h_i$ is given by

$$\frac{\partial \ell}{\partial \mathbf{w}^h_i} = \frac{\boldsymbol{\beta}^2_h e^{\mathbf{w}^h_i}}{\sum_{j=1}^k e^{\mathbf{w}^h_j}} \sum_{t=1}^{k!} \ell'_t \left( (1+\gamma)\mathbb{1}\{q_{k-h} = \pi_{(t,k-i)}\} - \left\langle e_{x_{l-h}}, \widetilde{\mathrm{I}}_k \, \mathbf{R}^h_t \right\rangle \right),$$

$$= \frac{\boldsymbol{\beta}^2_h e^{\alpha_h + \delta_h}}{e^{\alpha_h} + k - 1} \sum_{t=1}^{k!} \ell'_t \left( (1+\gamma)\mathbb{1}\{q_{k-h} = \pi_{(t,k-i)}\} - \left\langle e_{x_{l-h}}, \widetilde{\mathrm{I}}_k \, \mathbf{R}^h_t \right\rangle \right),$$

$$= \frac{\boldsymbol{\beta}^2_h e^{\alpha_h + \delta_h}}{e^{\alpha_h} + k - 1} \sum_{t=1}^{k!} \ell'_t \left( (1+\gamma)\mathbb{1}\{q_{k-h} = \pi_{(t,k-i)}\} - \frac{e^{\alpha_h} - 1}{e^{\alpha_h} + k - 1}(1+\gamma)\ell'_t \mathbb{1}\{q_{k-h} = \pi_{(t,k-h)}\} \right),$$

$$= (1+\gamma) \frac{\boldsymbol{\beta}^2_h e^{\alpha_h + \delta_h}}{e^{\alpha_h} + k - 1} \sum_{t=1}^{k!} \ell'_t \left( \mathbb{1}\{q_{k-h} = \pi_{(t,k-i)}\} - \frac{e^{\alpha_h} - 1}{e^{\alpha_h} + k - 1}\mathbb{1}\{q_{k-h} = \pi_{(t,k-h)}\} \right),$$

We introduce a notation where

$$\Delta_{h,i} = \sum_{t=1}^{k!} \ell'_t \left( \mathbb{1}\{q_{k-h} = \pi_{(t,k-i)}\} \right)$$

Using this notation, the gradients can be written as

$$\frac{\partial \ell}{\partial \boldsymbol{\beta}_h} = 2\boldsymbol{\beta}_h \frac{e^{\alpha_h} - 1}{e^{\alpha_h} + k - 1}(1+\gamma)\Delta_{h,h},$$

$$\frac{\partial \ell}{\partial \mathbf{w}^h_i} = (1+\gamma) \frac{\boldsymbol{\beta}^2_h e^{\alpha_h + \delta_h}}{e^{\alpha_h} + k - 1} \left( \Delta_{h,i} - \frac{e^{\alpha_h} - 1}{e^{\alpha_h} + k - 1}\Delta_{h,h} \right),$$

where $\Delta_{h,i} = \sum_{t=1}^{k!} \ell'_t \left( \mathbb{1}\{q_{k-h} = \pi_{(t,k-i)}\} \right)$. Substituting the expression of $\ell'_t$ we have

$$\Delta_{h,i} = \sum_{t=1}^{k!} -\mathbf{s}_t \left( \mathbb{1}\{r_t = r\} - \sum_{j=1}^{k!} \mathbf{s}_j \mathbb{1}\{r_j = r\} \right) \left( \mathbb{1}\{q_{k-h} = \pi_{(t,k-i)}\} \right),$$

$$= - \left[ \sum_{t=1}^{k!} \mathbf{s}_t \mathbb{1}\{r_t = r \wedge q_{k-h} = \pi_{(t,k-i)}\} - \left( \sum_{j=1}^{k!} \mathbf{s}_j \mathbb{1}\{r_j = r\} \right) \left( \sum_{t=1}^{k!} \mathbf{s}_t \left( \mathbb{1}\{q_{k-h} = \pi_{(t,k-i)}\} \right) \right) \right],$$

$$\Delta_{h,i} = - \left[ \sum_{t=1}^{k!} \mathbf{s}_t \mathbb{1}\{r_t = r \wedge q_{k-h} = \pi_{(t,k-i)}\} - \left( \sum_{j=1}^{k!} \mathbf{s}_j \mathbb{1}\{r_j = r\} \right) \left( \sum_{t=1}^{k!} \mathbf{s}_t \left( \mathbb{1}\{q_{k-h} = \pi_{(t,k-i)}\} \right) \right) \right].$$
(14)

where $s = \boldsymbol{\sigma}(\mathbf{s}')$. This completes the proof of the lemma.

$\square$

**Lemma C.3.** *For the parameter configuration* $\boldsymbol{\theta} = (\mathbf{w}^1, \mathbf{w}^2, \dots \mathbf{w}^k, \boldsymbol{\beta})$ *where* $\mathbf{w}^1 = \alpha_1 \mathbf{e}_h^1 + \delta_1 \mathbf{1}$ *and* $\boldsymbol{\beta}_h = 0$ *for* $h \neq 1$. *The following quantities are*

$$\mathbb{E}\Delta_{1,i} = \frac{1}{(e^{\gamma_1} + k - 1)^2} \left[ \frac{e^{\gamma_1}(|\mathcal{R}| - 1)}{(k-1)!|\mathcal{R}|} \right], \quad for\ i \neq 1.$$

$$\mathbb{E}\Delta_{1,1} = -(k-1)\mathbb{E}\Delta_{1,i} = -\frac{1}{(e^{\gamma_1} + k - 1)^2} \left[ \frac{e^{\gamma_1}(|\mathcal{R}| - 1)}{(k-2)!|\mathcal{R}|} \right].$$

*where* $\gamma_1 = (1 + \gamma)\boldsymbol{\beta}_1^2 \frac{e^{\alpha_1} - 1}{e^{\alpha_1} + k - 1}$.

*Proof.* Recall that $\mathbf{s}_t' = \gamma_1 \mathbb{1}\{q_{k-1} = \pi_{(t,k-1)}\}$ as $\gamma_h = 0$ for $h \neq 1$ and the softmax score is given by

$$\mathbf{s}_t = \frac{e^{\gamma_1 \mathbb{1}\{q_{k-1} = \pi_{(t,k-1)}\}}}{\sum_{j=1}^{k!} e^{\gamma_1 \mathbb{1}\{q_{k-1} = \pi_{(j,k-1)}\}}} = \frac{(e^{\gamma_1} - 1)\,\mathbb{1}\{q_{k-1} = \pi_{(t,k-1)}\} + 1}{e^{\gamma_1} + k - 1} \frac{1}{(k-1)!}.$$

First lets compute $\mathbb{E}\Delta_{1,i}$ for $i \neq 1$, Note that

$$\sum_{t=1}^{k!} \mathbf{s}_t \left( \mathbb{1}\{q_{k-1} = \pi_{(t,k-i)}\} \right) = \sum_{t=1}^{k!} \frac{(e^{\gamma_1} - 1)\,\mathbb{1}\{q_{k-1} = \pi_{(t,k-i)}\} + 1}{e^{\gamma_1} + k - 1} \frac{1}{(k-1)!} \left( \mathbb{1}\{q_{k-1} = \pi_{(t,k-i)}\} \right),$$

$$= \frac{1}{e^{\gamma_1} + k - 1}.$$

For $i = 1$, we have,

$$\sum_{t=1}^{k!} \mathbf{s}_t \left( \mathbb{1}\{q_{k-1} = \pi_{(t,k-i)}\} \right) = \sum_{t=1}^{k!} \frac{(e^{\gamma_1} - 1)\,\mathbb{1}\{q_{k-1} = \pi_{(t,k-i)}\} + 1}{e^{\gamma_1} + k - 1} \frac{1}{(k-1)!} \left( \mathbb{1}\{q_{k-1} = \pi_{(t,k-i)}\} \right),$$

$$= \frac{e^{\gamma_1}}{e^{\gamma_1} + k - 1}.$$

Now we can compute $\mathbb{E}\Delta_{1,i}$ for $i \neq 1$ as follows:

$$\mathbb{E}\Delta_{1,i} = -\mathbb{E} \left[ \sum_{t=1}^{k!} \mathbf{s}_t \mathbb{1}\{r_t = r \wedge q_{k-1} = \pi_{(t,k-i)}\} - \left( \sum_{j=1}^{k!} \mathbf{s}_j \mathbb{1}\{r_j = r\} \right) \left( \sum_{t=1}^{k!} \mathbf{s}_t \left( \mathbb{1}\{q_{k-1} = \pi_{(t,k-i)}\} \right) \right) \right],$$

$$= -\mathbb{E} \left[ \sum_{t=1}^{k!} \mathbf{s}_t \mathbb{1}\{r_t = r \wedge q_{k-1} = \pi_{(t,k-i)}\} \right] + \mathbb{E} \left[ \left( \sum_{j=1}^{k!} \mathbf{s}_j \mathbb{1}\{r_j = r\} \right) \frac{1}{e^{\gamma_1} + k - 1} \right],$$

For $i \neq 1$,

$$\mathbb{E} \left[ \sum_{t=1}^{k!} \mathbf{s}_t \mathbb{1}\{r_t = r \wedge q_{k-1} = \pi_{(t,k-i)}\} \right] = \frac{1}{e^{\gamma_1} + k - 1} \frac{1}{(k-1)!} \mathbb{E} \left[ \sum_{t=1}^{k!} \mathbb{1}\{r_t = r \wedge q_{k-1} = \pi_{(t,k-i)}\} \right],$$

$$= \frac{1}{e^{\gamma_1} + k - 1} \frac{1}{(k-1)!} \sum_{t=1}^{k!} \mathbb{P}(r_t = r \wedge q_{k-1} = \pi_{(t,k-i)}),$$

Note that for $i \neq 1$, $\mathbb{P}(r_t = r \wedge q_{k-1} = \pi_{(t,k-i)}) = \mathbb{P}(r_t = r)\mathbb{P}(q_{k-1} = \pi_{(t,k-i)})$ due to the independence as $q$ is not same as $\pi_t$. $\mathbb{P}(q_{k-1} = \pi_{(t,k-i)}) = \frac{1}{k}$ as $q_{k-1}$ is uniformly distributed over $[k]$. $\mathbb{P}(r_t = r) = \frac{1}{|\mathcal{R}|}$. So,

$$\mathbb{E}\left[\sum_{t=1}^{k!} \mathbf{s}_t \mathbb{1}\{r_t = r \wedge q_{k-1} = \pi_{(t,k-i)}\}\right] = \frac{1}{e^{\gamma_1} + k - 1} \frac{1}{(k-1)!} \sum_{t=1}^{k!} \frac{1}{k|\mathcal{R}|} = \frac{1}{e^{\gamma_1} + k - 1} \frac{1}{|\mathcal{R}|}.$$

Now for $i = 1$,

$$E\left[\sum_{t=1}^{k!} \mathbf{s}_t \mathbb{1}\{r_t = r \wedge q_{k-1} = \pi_{(t,k-1)}\}\right] = \frac{e^{\gamma_1}}{e^{\gamma_1} + k - 1} \frac{1}{(k-1)!} \mathbb{E}\left[\sum_{t=1}^{k!} \mathbb{1}\{r_t = r \wedge q_{k-1} = \pi_{(t,k-1)}\}\right],$$

Coming to the expectation term, now they are no longer independent.

$$\mathbb{E}\left[\sum_{t=1}^{k!} \mathbb{1}\{r_t = r \wedge q_{k-1} = \pi_{(t,k-1)}\}\right] = \sum_{t=1}^{k!} \mathbb{P}(r_t = r \wedge q_{k-1} = \pi_{(t,k-1)}),$$

$$= \sum_{t=1}^{k!} \mathbb{P}(r_t = r | q_{k-1} = \pi_{(t,k-1)})\mathbb{P}(q_{k-1} = \pi_{(t,k-1)}),$$

$$= \sum_{t=1}^{k!} \left[\mathbb{P}(r_t = r, q = \pi_t | q_{k-1} = \pi_{(t,k-1)}) + \mathbb{P}(r_t = r, q \neq \pi_t | q_{k-1} = \pi_{(t,k-1)})\right] \mathbb{P}(q_{k-1} = \pi_{(t,k-1)}),$$

$$\mathbb{P}(r_t = r, q = \pi_t | q_{k-1} = \pi_{(t,k-1)}) = \mathbb{P}(r_t = r | q = \pi_t, q_{k-1} = \pi_{(t,k-1)})\mathbb{P}(q = \pi_t | q_{k-1} = \pi_{(t,k-1)}),$$

$$= 1 \cdot \frac{1}{(k-1)!} = \frac{1}{(k-1)!}.$$

$$\mathbb{P}(r_t = r, q \neq \pi_t | q_{k-1} = \pi_{(t,k-1)}) = \mathbb{P}(r_t = r | q \neq \pi_t, q_{k-1} = \pi_{(t,k-1)})\mathbb{P}(q \neq \pi_t | q_{k-1} = \pi_{(t,k-1)}),$$

$$= \frac{1}{|\mathcal{R}|} \cdot \left[1 - \frac{1}{(k-1)!}\right]$$

Summming them up and substituting in the previous equation, we get,

$$\mathbb{E}\left[\sum_{t=1}^{k!} \mathbb{1}\{r_t = r \wedge q_{k-1} = \pi_{(t,k-1)}\}\right] = \sum_{t=1}^{k!} \left[\frac{1}{(k-1)!} + \frac{1}{|\mathcal{R}|} \cdot \left(1 - \frac{1}{(k-1)!}\right)\right] \frac{1}{k},$$

$$= k! \left[\frac{1}{(k-1)!} + \frac{1}{|\mathcal{R}|} \cdot \left(1 - \frac{1}{(k-1)!}\right)\right] \frac{1}{k},$$

$$= \frac{|\mathcal{R}| - 1}{|\mathcal{R}|} + \frac{(k-1)!}{|\mathcal{R}|}.$$

Substituting this back in the expectation term, we get,

$$E\left[\sum_{t=1}^{k!} \mathbf{s}_t \mathbb{1}\{r_t = r \wedge q_{k-1} = \pi_{(t,k-1)}\}\right] = \frac{e^{\gamma_1}}{e^{\gamma_1} + k - 1} \frac{1}{(k-1)!} \mathbb{E}\left[\sum_{t=1}^{k!} \mathbb{1}\{r_t = r \wedge q_{k-1} = \pi_{(t,k-1)}\}\right]$$

$$= \frac{e^{\gamma_1}}{e^{\gamma_1} + k - 1} \frac{1}{(k-1)!} \left[\frac{|\mathcal{R}| - 1}{|\mathcal{R}|} + \frac{(k-1)!}{|\mathcal{R}|}\right],$$

$$= \frac{e^{\gamma_1}}{e^{\gamma_1} + k - 1} \left[\frac{|\mathcal{R}| - 1}{(k-1)!|\mathcal{R}|}\right] + \frac{e^{\gamma_1}}{e^{\gamma_1} + k - 1} \frac{1}{|\mathcal{R}|}.$$

Recall the term for $i \neq i$, we have,

$$\mathbb{E}\left[\sum_{t=1}^{k!} \mathbf{s}_t \mathbb{1}\{r_t = r \wedge q_{k-1} = \pi_{(t,k-i)}\}\right] = \frac{1}{e^{\gamma_1} + k - 1} \frac{1}{(k-1)!} \sum_{t=1}^{k!} \frac{1}{k|\mathcal{R}|} = \frac{1}{e^{\gamma_1} + k - 1} \frac{1}{|\mathcal{R}|}.$$

Note that

$$\mathbb{E}\left[\sum_{t=1}^{k!} \mathbf{s}_t \mathbb{1}\{r_t = r\}\right] = \sum_{i=1}^{k} \mathbb{E}\left[\sum_{t=1}^{k!} \mathbf{s}_t \mathbb{1}\{r_t = r \wedge q_{k-1} = \pi_{(t,k-i)}\}\right],$$

$$= \frac{e^{\gamma_1}}{e^{\gamma_1} + k - 1}\left[\frac{|\mathcal{R}| - 1}{(k-1)!|\mathcal{R}|}\right] + \frac{1}{|\mathcal{R}|}.$$

Now for $i \neq 1$, we have,

$$\mathbb{E}\Delta_{1,i} = -\mathbb{E}\left[\sum_{t=1}^{k!} \mathbf{s}_t \mathbb{1}\{r_t = r \wedge q_{k-1} = \pi_{(t,k-i)}\} - \left(\sum_{j=1}^{k!} \mathbf{s}_j \mathbb{1}\{r_j = r\}\right)\frac{1}{e^{\gamma_1} + k - 1}\right],$$

$$= -\left[\frac{1}{e^{\gamma_1} + k - 1}\frac{1}{|\mathcal{R}|} - \left(\frac{e^{\gamma_1}}{e^{\gamma_1} + k - 1}\left[\frac{|\mathcal{R}| - 1}{(k-1)!|\mathcal{R}|}\right] + \frac{1}{|\mathcal{R}|}\right)\frac{1}{e^{\gamma_1} + k - 1}\right],$$

$$= \frac{1}{(e^{\gamma_1} + k - 1)^2}\left[\frac{e^{\gamma_1}(|\mathcal{R}| - 1)}{(k-1)!|\mathcal{R}|}\right].$$

Now for $i = 1$, we have,

$$\mathbb{E}\Delta_{1,1} = -\mathbb{E}\left[\sum_{t=1}^{k!} \mathbf{s}_t \mathbb{1}\{r_t = r \wedge q_{k-1} = \pi_{(t,k-1)}\} - \left(\sum_{j=1}^{k!} \mathbf{s}_j \mathbb{1}\{r_j = r\}\right)\frac{e^{\gamma_1}}{e^{\gamma_1} + k - 1}\right],$$

$$= -\left[\frac{e^{\gamma_1}}{e^{\gamma_1} + k - 1}\left[\frac{|\mathcal{R}| - 1}{(k-1)!|\mathcal{R}|}\right] + \frac{e^{\gamma_1}}{e^{\gamma_1} + k - 1}\frac{1}{|\mathcal{R}|} - \left(\frac{e^{\gamma_1}}{e^{\gamma_1} + k - 1}\left[\frac{|\mathcal{R}| - 1}{(k-1)!|\mathcal{R}|}\right] + \frac{1}{|\mathcal{R}|}\right)\frac{e^{\gamma_1}}{e^{\gamma_1} + k - 1}\right],$$

$$= -\left[\frac{e^{\gamma_1}}{e^{\gamma_1} + k - 1}\left[\frac{|\mathcal{R}| - 1}{(k-1)!|\mathcal{R}|}\right] - \frac{(e^{\gamma_1})^2}{(e^{\gamma_1} + k - 1)^2}\left[\frac{|\mathcal{R}| - 1}{(k-1)!|\mathcal{R}|}\right]\right],$$

$$= -\left[\frac{e^{\gamma_1}(k-1)}{(e^{\gamma_1} + k - 1)^2}\left[\frac{|\mathcal{R}| - 1}{(k-1)!|\mathcal{R}|}\right]\right] = -(k-1)\mathbb{E}\Delta_{1,i} \text{ for } i \neq 1.$$

$\square$

# D  TECHNICAL LEMMAS

**Gradient of the first layer.**  For both relative positional encoding, the gradient of the first layer can be computed as follows.

**Lemma D.1.** *Denote the forward pass of the first layer as follows,*

$$\mathbf{R}^h = \boldsymbol{\sigma}(\mathbf{A}^h)\,X, \quad \mathbf{R}_t^h = X^\top \boldsymbol{\sigma}(\mathbf{A}^h)_t = \sum_{i=0}^{t} \boldsymbol{\sigma}(\mathbf{A}^h)_{t,i}\,\mathsf{e}_{x_i}^{\mathcal{S}},$$

*The gradient of the embeddings with respect to the parameters $\mathbf{w}_i^h$ is given by*

$$\frac{\partial \mathbf{R}_t^h}{\partial \mathbf{w}_i^h} = \begin{cases} \dfrac{e^{\mathbf{w}_i^h}}{\sum_{j=0}^{t} e^{\mathbf{w}_j^h}}\left(\mathsf{e}_{x_{t-i}}^{\mathcal{S}} - \mathbf{R}_t^h\right) & \text{if } i \leqslant t, \\ 0 & \text{if } i > t, \end{cases}$$

*Proof.*

$$\mathbf{R}^h = \boldsymbol{\sigma}(\mathbf{A}^h)\,X,$$

$$\mathbf{R}_t^h = X^\top \boldsymbol{\sigma}(\mathbf{A}^h)_t = \sum_{i=0}^{t} \boldsymbol{\sigma}(\mathbf{A}^h)_{t,i}\,\mathsf{e}_{x_i}^{\mathcal{S}}$$

Recall that the matrix $\mathbf{A}^h$ is defined as follows,

$$
\mathbf{A}^h = \begin{bmatrix}
\mathbf{w}_0^h & 0 & 0 & \cdots & 0 \\
\mathbf{w}_1^h & \mathbf{w}_0^h & 0 & \cdots & 0 \\
\mathbf{w}_2^h & \mathbf{w}_1^h & \mathbf{w}_0^h & \cdots & 0 \\
\vdots & \vdots & \vdots & \ddots & \vdots \\
\mathbf{w}_{l-2}^h & \mathbf{w}_{l-3}^h & \cdots & \mathbf{w}_0^h & 0 \\
\mathbf{w}_{l-1}^h & \mathbf{w}_{l-2}^h & \mathbf{w}_{l-3}^h & \cdots & \mathbf{w}_0^h
\end{bmatrix}
$$

Now,

$$
\boldsymbol{\sigma}(\mathbf{A}^h)_{t,i} = \frac{e^{\mathbf{w}_{t-i}^h}}{\sum\limits_{j=0}^{t} e^{\mathbf{w}_j^h}},
$$

$$
\mathbf{R}_t^h = \sum_{i=0}^{t} \frac{e^{\mathbf{w}_{t-i}^h}}{\sum_{j=0}^{t} e^{\mathbf{w}_j^h}} \mathsf{e}_{x_i}^{\mathcal{S}} = \sum_{i=0}^{t} \frac{e^{\mathbf{w}_i^h}}{\sum_{j=0}^{t} e^{\mathbf{w}_j^h}} \mathsf{e}_{x_{t-i}}^{\mathcal{S}},
$$

$$
\text{For } i \leqslant t, \quad \frac{\partial \mathbf{R}_t^h}{\partial \mathbf{w}_i^h} = \frac{e^{\mathbf{w}_i^h}}{\sum_{j=0}^{t} e^{\mathbf{w}_j^h}} \mathsf{e}_{x_{t-i}}^{\mathcal{S}} - \frac{e^{\mathbf{w}_i^h}}{\left(\sum_{j=0}^{t} e^{\mathbf{w}_j^h}\right)^2} \sum_{j=1}^{t} e^{\mathbf{w}_j^h} \mathsf{e}_{x_{t-j}}^{\mathcal{S}},
$$

$$
= \frac{e^{\mathbf{w}_i^h}}{\sum_{j=0}^{t} e^{\mathbf{w}_j^h}} \left( \mathsf{e}_{x_{t-i}}^{\mathcal{S}} - \mathbf{R}_t^h \right)
$$

So, we have the result as follows,

$$
\frac{\partial \mathbf{R}_t^h}{\partial \mathbf{w}_i^h} = \begin{cases} \dfrac{e^{\mathbf{w}_i^h}}{\sum_{j=0}^{t} e^{\mathbf{w}_j^h}} \left( \mathsf{e}_{x_{t-i}}^{\mathcal{S}} - \mathbf{R}_t^h \right) & \text{if } i \leqslant t, \\ 0 & \text{if } i > t, \end{cases}
$$

$\square$

Now in the context of simplified transformer model with a fixed window we have the following

**Lemma D.2.** *Denote the forward pass of the first layer as follows,*

$$
\mathbf{R}^h = \boldsymbol{\sigma}(\mathbf{A}^h)\,X, \quad \mathbf{R}_t^h = X^\top \boldsymbol{\sigma}(\mathbf{A}^h)_t = \sum_{i=1}^{k} \boldsymbol{\sigma}(\mathbf{A}^h)_{t,t-i}\, e_{\pi_{(t,k-i)}},
$$

*The gradient of the embeddings with respect to the parameters $\mathbf{w}_i^h$ is given by*

$$
\frac{\partial \mathbf{R}_t^h}{\partial \mathbf{w}_i^h} = \frac{e^{\mathbf{w}_i^h}}{\sum\limits_{j=1}^{k} e^{\mathbf{w}_j^h}} \left( e_{\pi_{(t,k-i)}} - \mathbf{R}_t^h \right)
$$

*Proof.*

$$
\mathbf{R}^h = \boldsymbol{\sigma}(\mathbf{A}^h)\,X, \quad \mathbf{R}_t^h = X^\top \boldsymbol{\sigma}(\mathbf{A}^h)_t = \sum_{i=1}^{k} \boldsymbol{\sigma}(\mathbf{A}^h)_{t,t-i}\, e_{\pi_{(t,k-i)}},
$$

$$\sigma(\mathbf{A}^h)_{t,t-i} = \frac{e^{\mathbf{w}_i^h}}{\sum\limits_{j=1}^{k} e^{\mathbf{w}_j^h}},$$

$$\mathbf{R}_t^h = \sum_{i=1}^{k} \frac{e^{\mathbf{w}_i^h}}{\sum\limits_{j=1}^{k} e^{\mathbf{w}_j^h}} \, e_{\pi_{(t,k-i)}},$$

$$\text{For } 1 \leqslant i \leqslant k, \quad \frac{\partial \mathbf{R}_t^h}{\partial \mathbf{w}_i^h} = \frac{e^{\mathbf{w}_i^h}}{\sum\limits_{j=1}^{k} e^{\mathbf{w}_j^h}} \left( e_{\pi_{(t,k-i)}} - \mathbf{R}_t^h \right)$$

$\square$

## E    ADDITIONAL EXPERIMENTS

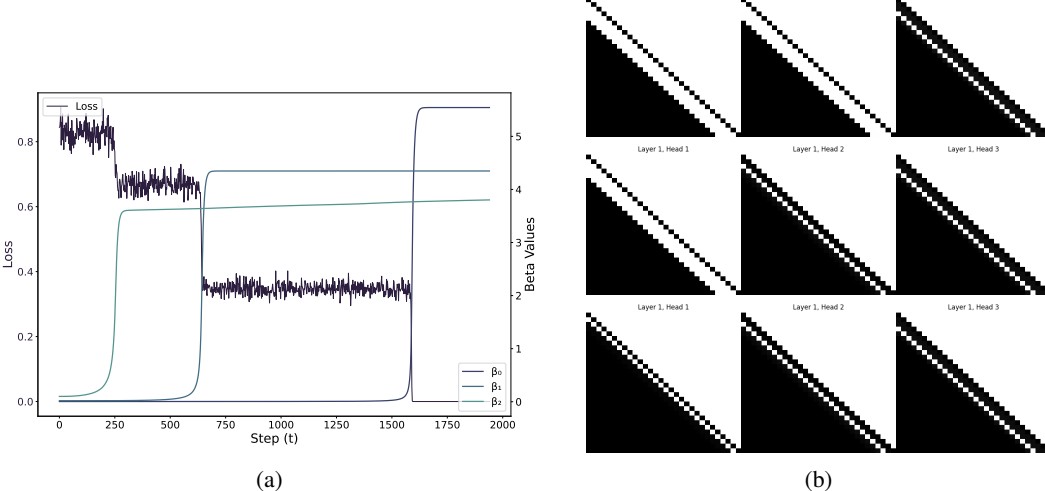

(a)                                                (b)

Figure 4: The left panel is shown in a and the right panel in b. Note that the $\beta$ values rise as the loss drops, and the attention patterns at times $800$, $1750$, and $4500$ demonstrate incremental learning. This is in the opposite order as they are initialized in the opposite order $\beta_3 > \beta_2 > \beta_1$

## F  ADDITIONAL EXPERIMENTS FOR REBUTTAL

We train a two–layer simplified transformer model and perform a series of ablation studies to examine the influence of different modeling assumptions. In particular, we evaluate the impact of assumptions **(A0)**, **(A1)**, **(A2)**, and **(A3)** by selectively relaxing them during training. Results under relaxed assumptions are shown in the accompanying plots; refer to the glossary below for a concise description of each assumption and its role in the analysis.

**Glossary.**  To quickly find the plots corresponding to different assumptions, refer to the following glossary:

- **Simplified model** — The baseline two-layer transformer model trained under all assumptions **(A0–A3)**. Please refer to Figure 5 and subsec. F.0.1.
- **Relaxing (A0) and** (A3)  — Experiments where the fixed window size and certain attention constraints only on the response tokens are removed. See Figure 6 and subsec. F.0.2.
- **Attention only transformer** — Further relaxations involving attention-only mechanisms without assumptions **A0–A4**. See Figure 7 and subsection F.0.3.
- **Full transformer architecture.**  We train a full transformer with multiple heads in the first layer and a single head in the second layer on a task of order 2. See Figure 8 and section F.0.4 for more details.
- **Without $\beta^2$ term** — Experiments investigating the role of the $\beta^2$ parameterization. See Figure 9, Figure 10 and subsection F.0.5.
- 

### F.0.1  TRAINING DYNAMICS ACROSS INITIALIZATION SCALES WITH THE SIMPLE TRANSFORMER

First, we will provide empirical evidence for the training dynamics of simplified two-layer transformer model under assumptions **A0-3** verifying the theoritical claims made in Theorem 3.1 and Theorem 3.2. Here $k = 4$ and we need a transformer with $k − 1 = 3$ heads in the first layer. We trained the Transformer on the cross-entropy loss using mini-batch SGD with a momentum parameter $\beta = 0.95$. See Figure 5 for the training dynamics and attention patterns. We rescale the time axis by the initialization scale $\epsilon$, i.e., $t \to t/\epsilon$ to highlight the characteristic timescale of parameter jumps. The trajectory converge to a limiting trajectory showing that the timescale of the jumps is $O(1/\epsilon)$.

Furthermore, we observe the conservation law between $\beta$ and $\alpha$ parameters in the right plot of Figure 5a. At the start of evolution $\beta \approx \alpha$, as the conservation states at the small scales and they grow in scale together as the loss abruptly decreases at the end of each stage. Finally, we observe the emergence of sparse attention and directional bias in Figure 5b. Note that first $\mathbf{w}_1^1$ of the head grows and all the remaining relative position parameters decay together confirming the theoretical prediction of Theorem 3.1. After the jump they do are stationary as shown in Theorem 3.2. In the next stage, $\mathbf{w}_1^2$ grows and the remaining relative position parameters ($\mathbf{w}_3^2, \mathbf{w}_4^2$) decay together. Note that $\mathbf{w}_1^2$ also decays which our theory does not capture this is due to finite scale effects.

### F.0.2  BEYOND ASSUMPTIONS A0 AND A3

In this section, we will present experimental results that relax assumptions **(A0)** and **(A3)**. Specifically, we will explore the effects of removing the fixed window size constraint and allowing for more general attention mechanisms. The results demonstrate that even without these assumptions, the model still exhibits key behaviors of incremental learning. In Figure 6, we experiment with $k = 4$ and due to removing assumption **(A3)**, we use $k = 4$ heads in the first layer. We trained the Transformer on the cross-entropy loss using mini-batch SGD with a momentum parameter $\beta = 0.95$. The attention patterns in Figure 6b show that the model still learns to focus on the relevant positions incrementally over time, similar to the behavior observed under the original assumptions. The loss and $\beta$ dynamics in Figure 6a further confirm that the training process remains stagewise and the abrupt change in loss is coupled with increase in the $\beta$ parameters.

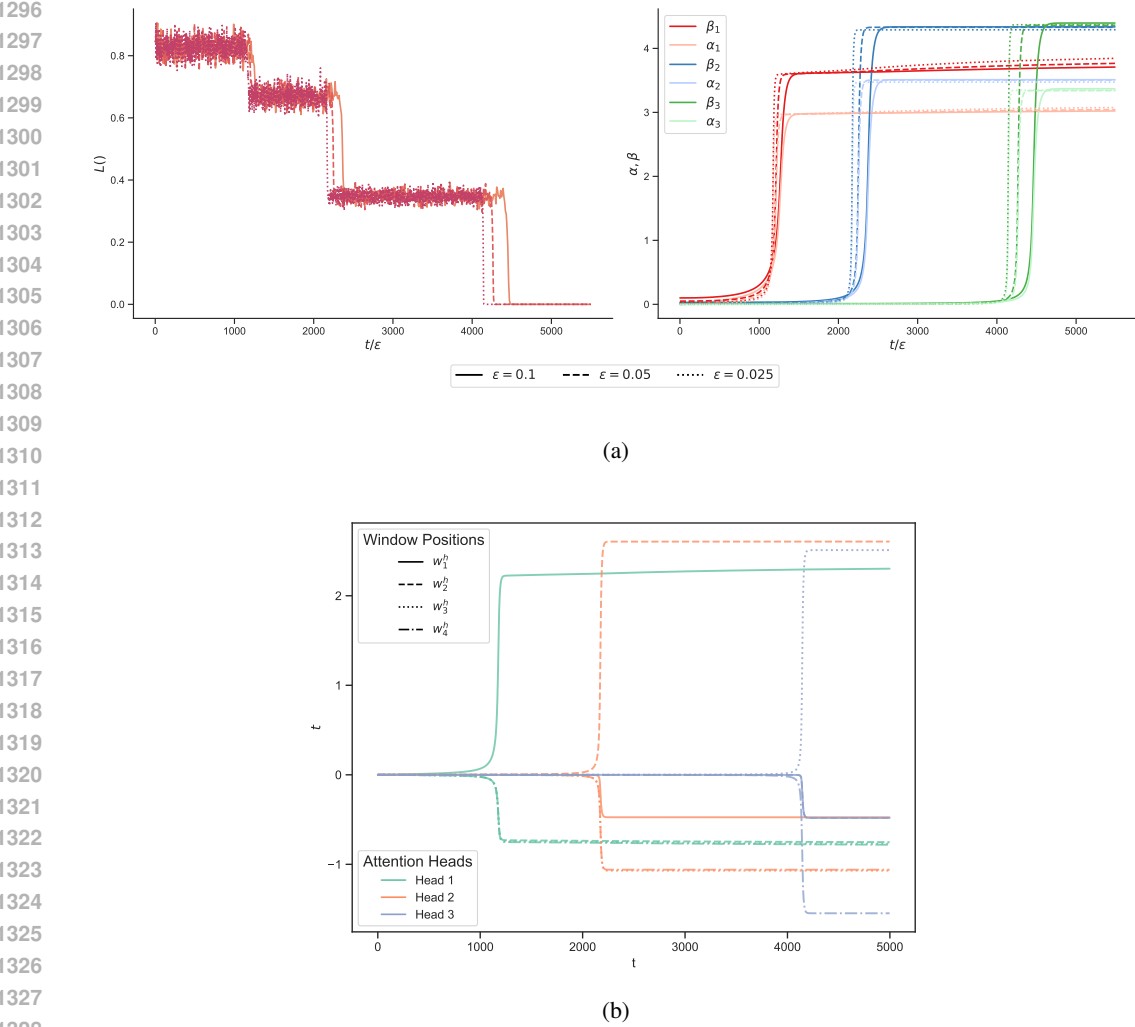

(a)

(b)

Figure 5: We train a **simplified two-layer transformer** using **cross-entropy loss** under the assumptions **(A0–A3)**. In **(a)**, we initialize the parameter vector $\boldsymbol{\beta}$ as $\boldsymbol{\beta} = \epsilon(\beta_1, \beta_2, \beta_3)$ where $\beta_1 > \beta_2 > \beta_3$, for different values of $\epsilon$. The dynamics are plotted with respect to a **rescaled time** $t/\epsilon$, where the convergence of the trajectories reveals the characteristic timescale for the parameter jumps is $O(1/\epsilon)$, with smaller $\epsilon$ leading to sharper jumps. The left side of **(a)** shows the loss convergence, and the right side illustrates the coupled evolution of $\boldsymbol{\beta}$ and $\boldsymbol{\alpha}$'s, which verifies the **conservation law** at play. Finally, **(b)** plots the relative position parameters across the various attention heads for $\epsilon = 0.025$, confirming both the **directional bias** and the emergence of **sparse attention**.

In this section, we will present the empirical evidence without the assumptions **A0-4** of the attention only transformer model with multi-headed disentangled attention defined in Section 2.2. We trained the Transformer on the cross-entropy loss using mini-batch SGD with a momentum parameter $\beta = 0.95$. The experiments demonstrate that without these assumptions, the model exhibits the incremental learning behaviour. See Figure 7 for a comprehensive analysis of multiple attention heads and their dynamics.

More importantly, in Figure 7c, we analyze the $\mathbf{Q}^\top \mathbf{K}$ matrices for the parameters in the second layer. We observe that the $\mathbf{Q}^\top \mathbf{K}$ matrices maintain a block structure. Each block seems to have a form where the diagonal elements are positive and the off-diagonal elements are negative, so it is a scaled version of $\widetilde{\mathrm{I}}_k$ with a positive scaling factor. Note that it is the same $\widetilde{\mathrm{I}}_k \in \mathbb{R}^{|\mathcal{S}| \times |\mathcal{S}|}$ where the first $k \times k$ block is given by $\widetilde{\mathrm{I}}_{:k,:k} = \mathrm{I}_k - \gamma \mathbf{1}_k \mathbf{1}_k^\top$ defined in the subsection 2.3. Let $\mathrm{r}_g^{(h)}$ be the output of head $h$ for the $g^{th}$ token. Then, with the block structure the presoftmax attention scores after the

**Beyond Assumptions A0 and A3**

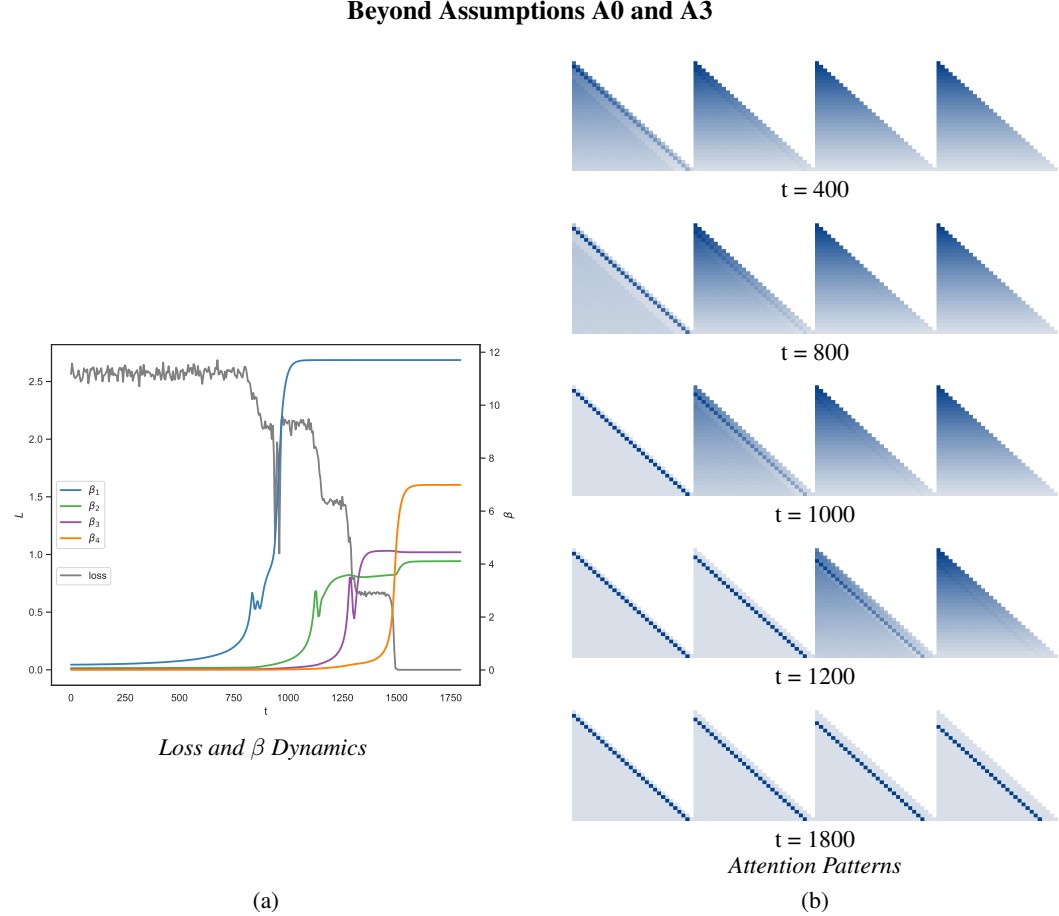

*Loss and $\beta$ Dynamics*

(a)

*Attention Patterns*

(b)

Figure 6: **Without Assumptions A0 and A3.** The left panel shows the loss and $\beta$ parameter dynamics. The right panel displays attention patterns at times 400, 800, 1000, 1200, and 1800 demonstrating incremental learning of the model.

second layer are given by

$$
\widetilde{\mathbf{s}}_t = \sum_{g,h=0}^{k} \left[\mathbf{R}_{l-1}^g\right]^\top \left[\mathbf{Q}^\top \mathbf{K}\right]_{gh} \left[\mathbf{R}_t^h\right],
$$

$$
\approx \sum_{h=1}^{k} \left[\beta_{h,h} \left[\mathbf{R}_{l-1}^h\right]^\top \widetilde{\mathbf{I}}_k \left[\mathbf{R}_t^h\right] - \sum_{g=0,g\neq h}^{k-1} \beta_{g,h} \left[\mathbf{R}_{l-1}^h\right]^\top \widetilde{\mathbf{I}}_k \left[\mathbf{R}_t^g\right]\right],
$$

$$
= \sum_{h=1}^{k} \left[2\beta_{h,h} \left[\mathbf{R}_{l-1}^h\right]^\top \widetilde{\mathbf{I}}_k \left[\mathbf{R}_t^h\right] - \sum_{g=0}^{k-1} \beta_{g,h} \left[\mathbf{R}_{l-1}^g\right]^\top \widetilde{\mathbf{I}}_k \left[\mathbf{R}_t^h\right]\right],
$$

where $\beta_{h,g}$ are some positive constant as seen in the Figure 7c.As $\sum_{g=0}^{k-1} \beta_{g,h} \left[\mathbf{R}_{l-1}^g\right]^\top \widetilde{\mathbf{I}}_k \left[\mathbf{R}_t^h\right]$ is approximately constant addition for all $t$ and hence diagonally dominated just by the diagonal terms and this justify the assumption (A2).

### F.0.4 FULL TRANSFORMER MODEL

**Experimental Setting.** In this section, we present experimental results for the full transformer model without the simplifying assumptions used in our theoretical analysis. We consider the task with $k = 2$ and vocabulary size of 6. For this task, we train a transformer with 2 heads in the first layer and 1 head in the second layer. The transformer uses relative positional encoding, has MLP

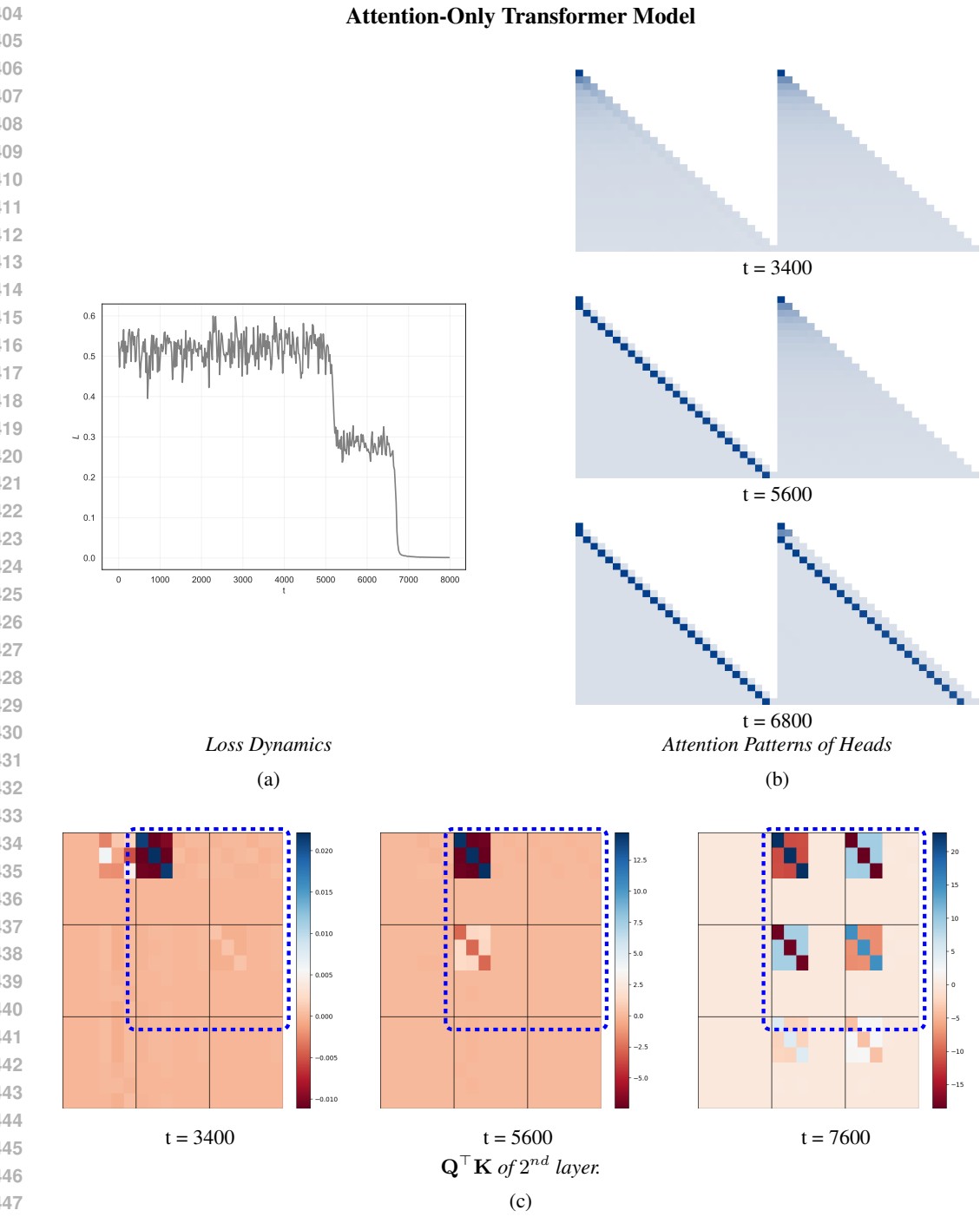

Figure 7: **Dynamics of attention-only transformer model.** **(a)** Loss dynamics during training. **(b)** Attention patterns for Head Attention 0 at times 3400, 5600, and 6800 exhibiting incremental learning **(c)** We consider the dynamics of $\mathbf{Q}^\top\mathbf{K}$ for the parameters in layer 2, we see that it maintains a block diagonal structure very close to the parameterization with assumption (A2).

layers, uses layernorm and skip connection. We train on the standard cross-entropy loss using Adam and a constant learning rate of 0.003. We initialize all the parameters with a standard Kaiming initialization.

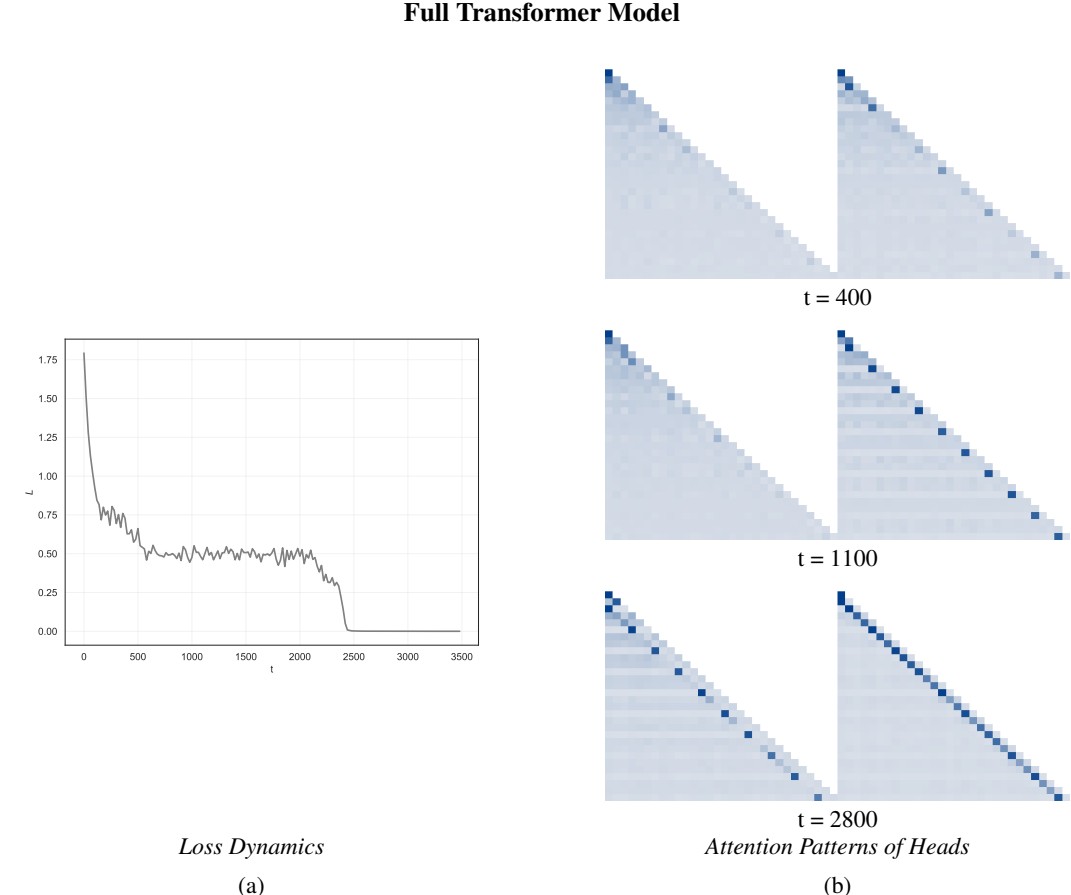

**Full Transformer Model**

*Loss Dynamics*

(a)

*Attention Patterns of Heads*

(b)

Figure 8: **Dynamics of full transformer model.** **(a)** Loss dynamics during training on task with order is 2, i.e., $k = 2$. **(b)** Attention patterns for Head Attention 0 at times 400, 1100, and 2800 exhibiting incremental learning. The heads at $t = 400$ do not attend to any particular tokens, while in the **second stage** the second head at time $t = 1100$ learns to attend to the previous token. Note that $k = 2$ and every 3rd token is a response token and it learns to attend its previous token. In the **final stage**, the first head learns to attend $(-2)$-token (at the response tokens) and solves the task.

Despite the additional complexity compared to the simplified model, Figure 8 demonstrates that the model still exhibits stage-wise learning behavior with clear transitions in the loss curves. The attention patterns of the heads show the incremental learning behaviour. Note that $k = 2$ and the response tokens appears with a periodicity of 3. At time $t = 400$ amdist a small plateau, the heads do not attend to any particular tokens. At $t = 1100$, second head becomes active and at response token (every 3rd token) learns to attend to the *previous* token. At $t = 2800$, the first head learns to attend to the $(-2)$-token, and each response token attends to their corresponding $(-2)$-token.

### F.0.5 POSITIVITY AND PARAMETERIZATION OF SECOND LAYER

In this subsection, we investigate the role of the $\beta^2$ parameterization by training the model without this term, using only the standard $\beta$ parameters. This investigation helps validate the robustness of the incremental learning phenomenon and provides additional insights into the role of parameter positivity constraints. See Figure 10 for the training dynamics with this alternative parameterization and a negative initialization. The remining of the experimental setting remains the same as in Section F.0.1.

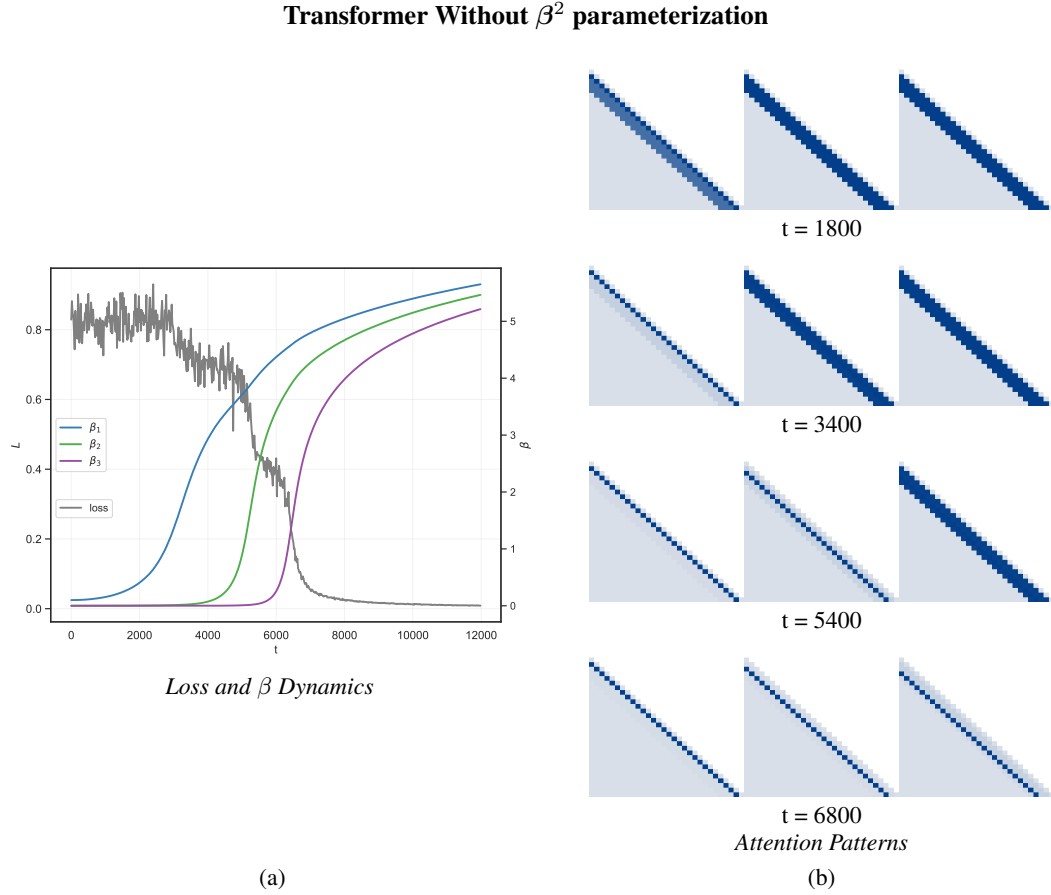

**Transformer Without $\beta^2$ parameterization**

*Loss and $\beta$ Dynamics*

(a)

t = 1800

t = 3400

t = 5400

t = 6800

*Attention Patterns*

(b)

Figure 9: The left panel shows the loss and $\beta$ parameter dynamics. The right panel displays attention patterns at times 1800, 3400, 5000, and 6800 demonstrating incremental learning without the $\beta^2$ term.

### F.0.6 INCOMPLETE PERMUTATIONS

In this section, we investigate the model's behavior when trained on incomplete permutations, where not all possible permutations are present in the training data. This variant tests the robustness of the incremental learning phenomenon to variations in the task structure. Figure 11 shows the loss and $\beta$ dynamics alongside the evolving attention patterns at different training stages. The results demonstrate that even with incomplete permutations, the model exhibits stage-wise learning behavior with clear phase transitions in loss and parameter values. Here we choose a task of order 4 and the context contains 12 permutations instead of the full set of 24 permutations. The remining of the experimental setting remains the same as in Section F.0.1.

**Training Dynamics with Negative Beta Parameters**

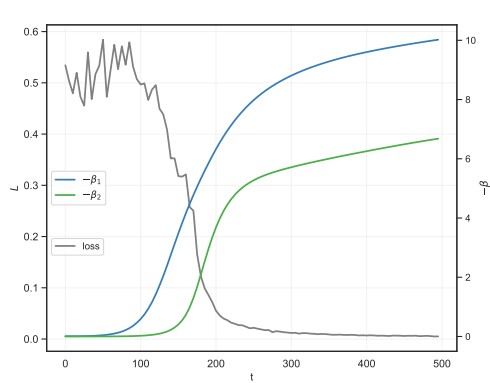

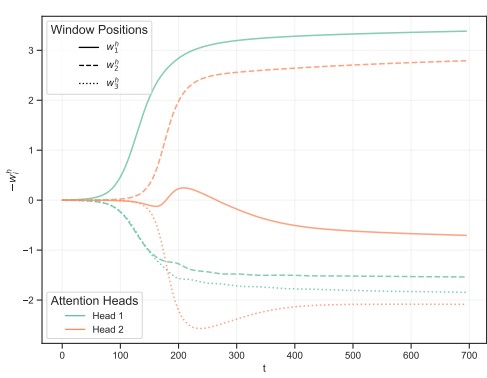

(a) Loss and negative beta parameters over time

(b) Weights of the window (attention head parameters) over time

Figure 10: **Training dynamics with negative $\beta$ parameterization. (a)** Loss dynamics (gray) alongside the negative beta parameters ($-\beta_i$, colored lines) showing their evolution during training. **(b)** The negative values of the fixed window weights ($-\mathbf{w}_i^h$) for each attention head and position. The plot shows convergence in the case of initialization with negative $\beta$'s.

**Incomplete Permutations**

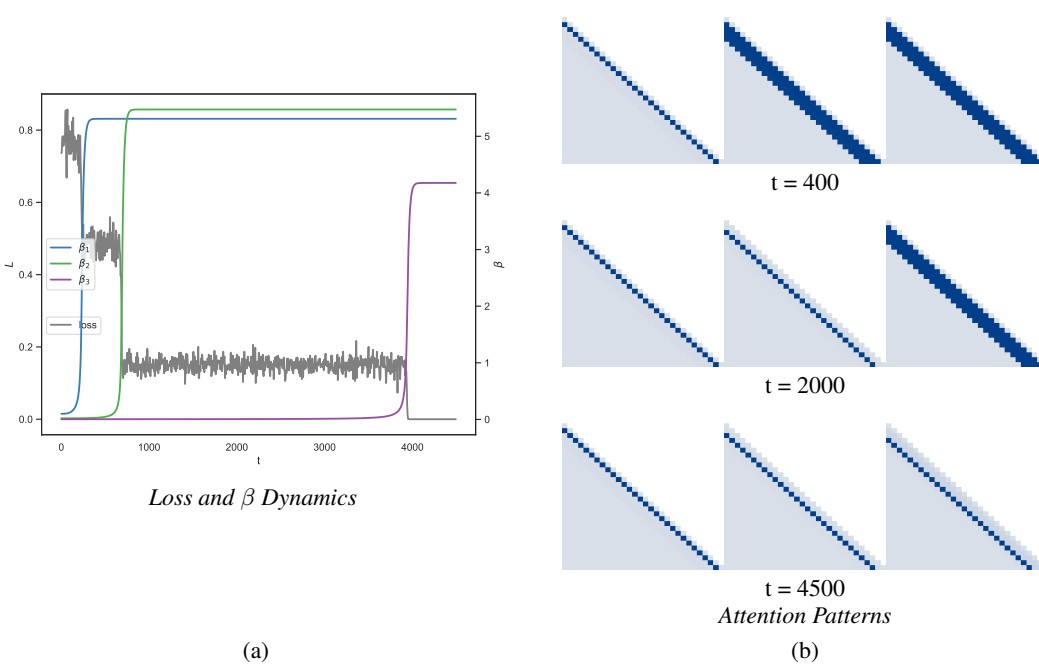

*Loss and $\beta$ Dynamics*

t = 400

t = 2000

t = 4500
*Attention Patterns*

(a)

(b)

Figure 11: **Incomplete Permutations.** The left panel shows the loss and $\beta$ parameter dynamics. The right panel displays attention patterns at times 400, 2000, and 4500 demonstrating how the model learns without all permutation present in the prompt.

