# OpenReview forum: "Incremental Learning in Transformers for In-Context Associative Recall"
_ICLR.cc/2026/Conference — Submitted to ICLR 2026_

### Official Review · Reviewer_g8Xr · 2025-10-26

**Soundness:** 3
**Presentation:** 2
**Contribution:** 2
**Rating:** 2
**Confidence:** 3

**Summary:**

The paper investigates how a small, two-layer attention-only transformer learns an n-gram recall in-context learning (ICL) task, where the model must retrieve the correct output by matching the final \(n=4\) tokens of a cue seen earlier in the same sequence. Building on previously proposed attention-based solutions for n-gram recall (Varré et al.), the paper constructs a minimal model that implements those solutions. This model admits a fully analytic treatment under gradient flow and captures the learning behavior observed in practice. Experiments training the model are consistent with the analytic predictions.

**Strengths:**

1. The motivation for introducing a minimal transformer model is clear. The proposed minimal model capture n-gram recall ICL solution.

2. The minimal transformer model is fully analytic under gradient flow and yields closed-form training dynamics. Analyzing these dynamics reveals a conservation law, initialization dependence, and phase ordering

**Weaknesses:**

1. The abstracted transformer architecture may be too minimal. While recent work aims to reduce complexity without losing fidelity, it remains unclear how much intuition from this abstraction transfers to real LM systems.

2. The focus is limited to n-gram ICL, a setting already known to work as described; the incremental insights appear modest.

3. Experiments are conducted in a specific setting without varying key factors such as \(N\) (the n-gram length) or architectural choices.

**Questions:**

1. Ablations (A1--A4): How do ablations of A1, A2, A3, and A4 affect empirical training outcomes? Does the phenomenology seen in the full simple model persist under these ablations?

2. Beyond n-gram ICL: How can the findings be extended to tasks outside n-gram recall? In particular, some classes of algorithmic ICL tasks (e.g., linear regression) have known transformer implementations (e.g., Lu et al., 2025). How can the paper’s findings be generalized to that class?

3. Timing of transitions: Does the model predict the time points of performance ``jumps''? If so, do these predictions match experimental results?

4. Initializations: Can you demonstrate how varying initialization affects the results in Figure 3?

References
[1] Varre, A., Yüce, G., Flammarion, N. “Learning In-context n-grams with Transformers: Sub-n-grams Are Near-stationary Points.” 2025

[2] Lu, Yue M., et al. "Asymptotic theory of in-context learning by linear attention." Proceedings of the National Academy of Sciences 122.28 (2025): e2502599122.

---

> ### Author Response · Authors · 2025-12-03
> **Rebuttal by Authors**
>
> We thank the reviewer for their questions which we address below.
>
> > **The focus is limited to n-gram ICL, a setting already known to work as described; the incremental insights appear modest**
>
> Besides incremental learning, just considering the case of learning induction head of order 1, our analysis studies a two layer transformer architecture while training both the layers (unlike layerwise training) and this in our opinion is also a valuable contribution.
> We emphasize that all prior analyses of n-gram ICL rely on layerwise or partially frozen training. Our work is the first to study training of a two-layer transformer, where both layers are trained jointly. This setting is much closer to practical transformer training and is technically more demanding, as it requires tracking the coupled dynamics across layers. Thus our contribution goes beyond the existing layerwise analyses and provides genuinely new insight into ICL in realistically trained models.
>
> > **Ablations and phenomenology**
>
> We refer to the **general response** and newly added section F in the appendix for discussion regarding ablation of the experiments.
>
> > **Generalization to Regression**
>
> Yes the observation of heads activating sequentially does happen in the case of regression. We refer to [1] where the heads in **linear-attention-only** transformer learn incrementally the principal components (see Fig. 5 therein). With softmax and multiheaded attention, [2] studied the convergence for  multi-task linear regression. It is shown that the attention heads where each head is specialized to a specific task are emerged after a warmup phase. However, in [2], all the heads emerge simultaneously due to the task-wise homogenity assumption hence incremental learning is not captured.
>
> [1] Alternating Gradient Flows: A Theory of Feature Learning in Two-layer Neural Networks, Kunin et. al. 2025
>
> [2] Training Dynamics of Multi-Head Softmax Attention  for In-Context Learning: Emergence, Convergence,  and Optimality, Chen et. al. 2024.
>
> >  **Timings of the further jumps**
>
> Yes for a well seperated initialization (e.g., $\beta_i = \epsilon^{i}$), the timings of the $i^{th}$ jump is $O(1 / \epsilon^{i} )$. The jump times in such cases are well seperated and a bound on the jump times can be given. For a  initialization $\epsilon (\beta_1, \beta_2 , \ldots )$ the activations are of order $O(1/\epsilon)$ as seen from Figure 5 in appendix, and given by  Theorem 3.1.  However,  the precise constants are harder to compute.
>
> > **Varying Initializations**
>
> Our analysis in the Appendix reveals key dependencies on initialization. Figure 5 demonstrates that decreasing the initialization scale leads to sharper jumps in the learning trajectories. Furthermore, Figure 4 illustrates that the initial order of the $\beta'$ parameters dictates the sequence of learning. Specifically, the self-attention head learning to attend to the $3^\text{rd}$ token from the last is acquired first, followed by the $2^\text{nd}$ token, and finally the $1^\text{st}$ token from the last.

---

### Official Review · Reviewer_XNc8 · 2025-10-27

**Soundness:** 2
**Presentation:** 3
**Contribution:** 3
**Rating:** 6
**Confidence:** 2

**Summary:**

The paper analyses how in-context learning circuits (e.g. induction heads) emerge during training in transformers.
To investigate this, the authors design:
* a controlled in-context associative recall task where the model must retrieve the correct response to a query by matching that query to an earlier key-value pair in the same prompt, and
* a heavily simplified 2-layer attention-only transformer where most parameters are fixed and only a few scalar/logit parameters are learned.

In this setting, they derive continuous-time gradient flow dynamics and prove that learning proceeds in stages: the model sits on plateaus where a partial recall circuit exists (e.g. can match only part of the query), then undergoes sharp “jumps” to a more complete recall mechanism. They link these jumps to saddle escape and show a conserved quantity that couples parameters across layers.
They further argue that which attention head “activates” first is determined by initialization scales, implying that heads specialize sequentially rather than all at once. Small experiments on order-4 recall show plateau to jump loss curves and sequential head specialization consistent with the theory.

**Strengths:**

1. Mechanistic training dynamics.
The work gives a concrete dynamical story for how in-context recall circuits form over time, including why capabilities appear in sharp jumps instead of smoothly.

2. Elegant task/model co-design.
The associative recall task and the simplified transformer still capture key ingredients of induction-style retrieval (causal attention, multi-head structure), but are analysable enough to get closed-form ODEs and a conservation law.

3. Theory explains qualitative phenomena people see in practice.
The staged plateaus, sudden capability jumps, and head specialisation order are all things observed in real small transformers; here we get a mechanistic account (ordered head activation driven by initialization scale, long dwell times near saddles).

4. Potential practical implication.
The result that init scales $\beta_h$ bias which head learns first suggests we can steer emergent specialization and maybe training curricula just by tuning initialization, which is actionable.

**Weaknesses:**

1. Gap to realistic models.
The theoretical guarantees rely on a very stylised transformer: fixed value structure, no MLPs, one-hot tokens, continuous-time gradient flow on full-population loss, etc. It's unclear how directly the results carry over to standard GPT-style models trained with SGD on natural data. The paper argues informally that the story should generalize, but does not really show it.

2. Light empirical support.
Experiments are on a tiny recall task with tiny vocab, and are mostly qualitative (plots of plateau/jump behavior, sequential $\beta_h$  growth). There’s little quantitative matching between the theoretical predictions (e.g. plateau duration scaling, ordering of head activation by init magnitude) and actual measured numbers.

3. SGD / noise is under-discussed.
The analysis assumes noiseless gradient flow. In practice, SGD noise helps models escape saddles. If saddle escape timing is central to the staged-learning story, then we need at least a discussion (or small ablation) of how noise affects plateau lengths and ordering.

4. Accessibility / reproducibility.
Some of the most interesting claims (like the conservation law and the staged head activation sequence) depend on fairly dense math and on training details that aren’t fully specified in the experimental section. This makes it harder for a broad ICLR audience to verify or reproduce.

**Questions:**

1. Generalization to standard transformers:
If you allow a more realistic transformer (trainable value matrices, MLPs, standard CE loss, SGD noise), do you still see the same staged head activation and ordered specialization? Have you run even small ablations in that more general setting?

2. Effect of SGD noise:
Your analysis is in deterministic gradient flow on the population loss. In actual training, SGD noise is often viewed as the mechanism that helps leave plateaus. Do you expect noise to (a) only change the timing of the jumps, or (b) potentially reorder which head “wins” first?

3. Quantitative match to theory:
You argue that the model dwells near partial circuits for long periods and then jumps. Can you report measured plateau durations vs. the theoretically predicted scaling (e.g. O(1/ϵ)) and show how close they are?

4. Scaling the task:
The associative recall task is “clean”: the answer always appears in the prompt, there’s exactly one correct continuation, and there’s no ambiguity. How do the dynamics change if the model has to generalize from incomplete evidence or noisy matches (i.e. more like natural text)?

5. Conservation law intuition:
The conserved quantity tying together first- and second-layer parameters is one of the most interesting parts of the paper. Can you give more geometric or mechanistic intuition for what it “means” operationally, in a way that a practitioner could try to measure in a non-simplified model?

---

> ### Author Response · Authors · 2025-12-03
> **Rebuttal by Authors**
>
> We thank the reviewer for their positive assessment of the work and point to the general response to extend the experiments beyond simple architectures.
>
>
> > **Gap to realistic models and generalization to standard transformers**
>
> We train a two layer transformer with MLPs, learnable relative positional embeddings, value matrices and layer normalization with mini-batch Adam on synthetic data. In Figure 8, we recover the incremental learning phenomenon where an head attending to the previous tokens emerge after each stage.
>
>
> > **Quantitative match to theory**
>
> We refer to Figure 5 in the appendix, where we show a quantitative match to **timescales, the conservation law in action and also the directional bias**.
>
>
> > **SGD noise**
>
> Yes the works considers the gradient flow on the population and do not deal with the stochastic nature of the gradients. To do so, one has to carefully study the martingale concentration, as done in the works on feature learning with MLPs[1] and determine the timescale of learning with SGD. We leave it for future work. However, empirically, with small batch sizes (e.g., b=128), we do observe incremental dynamics, indicating that tracking the population-loss dynamics captures some qualitative behavior of mini-batch SGD. Regarding the SGD noise, SGD noise when with small step-sizes to closely track the gradient flow, however with large stepsizes and small mini-batches they can potentially impact the order of learning and the espace time from the saddles. We will add this discussion to the revision of the paper.
>
> [1] Ben Arous, G., Gheissari, R., & Jagannath, A. (2021). Online stochastic gradient descent on non-convex losses from high-dimensional inference
>
>
> > **Scaling the task**
>
> Thank you for this interesting question. We designed a clean and controlled setting that enables us to derive the learning dynamics in closed form while still faithfully capturing the key phenomenology. In the one-hot token setting, the effect of noise tokens can be analyzed by projecting the dynamics onto the subspace orthogonal to the span of the noise tokens. However, extending our approach to handle partial matches is not straightforward: in our setup, response tokens are sampled independently, and partial matches do not induce any correlation. Exploring distributions in which partial matches produce correlated responses would be an interesting direction for future work.
>
>
> >  **Intution for conservation law**
>
> The conservation property is based on the existence of **symmetries** across the layers of the model. Specifically, the influence of the per-layer parameters $\beta_i$ and $\alpha_i$ on the final outcome is mediated **exclusively** through the composite parameter $\gamma_i$, defined in Equation 13 in the paper as:
>
> $$\gamma_i = \beta_i^2 \frac{e^{\alpha_i}}{e^{\alpha_i} + k - 1}$$
>
> Since the final outcome depends only on $\gamma_i$, any transformation of the primary parameters $(\beta_i, \alpha_i)$ that leaves $\gamma_i$ **invariant** constitutes a **symmetry** of the system. According to **Noether's Theorem**, every continuous symmetry of the action of a physical system corresponds to a conserved quantity or conservation law.

---

### Official Review · Reviewer_jH7v · 2025-11-01

**Soundness:** 3
**Presentation:** 4
**Contribution:** 3
**Rating:** 8
**Confidence:** 3

**Summary:**

This paper focuses on understanding the dynamics of emergence of mechanisms for in-context tasks. In particular, they characterize the emergence of circuits for an in-context recall task in a two-layer Transformer, and demonstrate that shape/scale of initialization can influence the timescales on which circuits are learned.

**Strengths:**

* The "methodological" contribution of the paper, i.e. understanding how internal mechanisms are influenced by optimization pressure, is extremely strong and very valuable, and will I think encourage interesting discussion.
* The work is theoretically and methodologically sound.
* The paper is very well written, figures are super helpful.

**Weaknesses:**

* I think (as with any work of this kind) there are some significant limitations in terms of realism of the setting: e.g. the paper focuses on a very small Transformer, most of the analysis is done in the vanishing scale of initialization, the synthetic task might not generalize to things we actually care about.

**Questions:**

See weaknesses.

---

> ### Author Response · Authors · 2025-12-03
> **Rebuttal by authors**
>
> We thank the reviewer for their positive assessment of our work. We direct the reviewer to the **General Response** and section F in the appendix, which details the extended experiments—including the ablation studies—that show the observed phenomenology is consistently maintained across a broader range of models, including the **standard** Transformer architecture.

---

### Official Review · Reviewer_UrhB · 2025-11-03

**Soundness:** 3
**Presentation:** 4
**Contribution:** 3
**Rating:** 2
**Confidence:** 3

**Summary:**

In this paper, the authors consider an in-context associative recall task trained using a simplified two-layer attention-only network. The main contribution is a quantitative description of the learning dynamics based on key parameters, which include the positional bias and diagonal elements of the key-query matrices.

**Strengths:**

The paper makes a series of simplifications to obtain insight into the training dynamics of a transformer. The simplifications are phenomenology-driven: by considering the key parameters that make up the sub-circuit that performs associative recall, they obtain a simplified description of the training dynamics that yet captures most elements of sub-circuit formation. I believe this is a strength of the paper -- the analysis provides some insight in scenarios where an exact solution cannot be derived.

**Weaknesses:**

In my opinion, the paper has three major weaknesses:

First, while the authors derive a series of mathematical expressions, none of these calculations are quantitatively tested in numerical simulations (except for a qualitative one in Figure 3). This seems like a missed opportunity to connect theory to experiment, which is noteworthy here because the simulations are not computationally expensive and the authors presumably already have an implementation (as evidenced by Figure 3, 4). Moreover, none of the details of the experiments are presented, which makes it impossible to interpret Figure 3,4. It is not clear whether the experiments were performed using a standard two-layer model or the simplified model. Without these experiments, it is impossible to say whether the theory the authors have written down using the simplified model accurately captures the dynamics of a full transformer model. Are there specific predictions that the theory offers which can be tested using numerics?

Second, there is a key positivity assumption made in assumption A1 (page 5 bottom). Here the query-key product is expressed in terms of \beta^2 rather than \beta. However, there is no constraint in a full transformer that imposes positivity of the diagonal entries, so it is unclear why one should assume positivity in the simplified model. How does relaxing the positivity assumption on \beta^2 change the analysis?
This is an important point, because if \beta^2 were replaced with \beta, one would find a saddle point close to initialization: this is apparent when one writes the expression for the ODE for \beta_1 (Theorem 3.1) in terms of \beta_1^2 rather than \beta_1. That is, the dynamics will flow to different basins depending on how \beta and \alpha are initialized. A similar phenomenon has been shown to occur in previous analyses. For example, see eqns 7-9 in https://openreview.net/forum?id=INyi7qUdjZ, who also derive the training dynamics of the induction head circuit in terms of effective parameters. In that setting, the saddle point is absent when one takes into account randomness in sampling input examples.

Third, the authors seem to be assuming that all (k+1)! permutations are shown in the context. This is an unrealistic assumption. I understand that this simplifies the analysis, but as pointed above, the randomness in sampling an input can indeed matter. It is unclear how this assumption will impact the generality of the results.

**Questions:**

Please see the questions raised in the weaknesses section above. In my opinion, the first point regarding the lack of numerical tests is sufficiently serious to warrant rejection -- at this stage, the theory has no empirical grounds on which it has firm footing, though this is of course not unresolvable.

---

> ### Author Response · Authors · 2025-12-03
> **Rebuttal by Authors**
>
> We thank the reviewer for the insightful remarks. We reply to their questions below.
>
>
> > **Positivity assumption and parameterizing as $\beta$**
>
> We thank the reviewer for raising this point which we will include in the revised version of the paper.
>
> **Negative $\\beta'$s**. The positivity of **$\beta$ is not required in our case** unlike Nyguen and Reddy 2025. This is due to the fact that $\beta_i <0$ and $\alpha_i <0$ is also a  solution to the problem we study. Consider the case of parameterization with $\beta$ instead of $\beta^2$. The presoftmax scores in second layer for token $t$ are given by (Equation 13 modified accordingly)
> $$\begin{align*}
> s_t^{'} = \sum_{h=1}^{k} \gamma\_{h} \mathbf{1} \\{ q_{k-h} =\pi_{t, k-h} \\}
> \end{align*}$$
> where $\gamma_h = \beta_h \frac{e^{\alpha_h} - 1}{e^{\alpha_h} + k -1}$, so $\alpha_h, \beta_h < 0$ also ensures that the $\gamma_h$ is positive and the desired solution is eventually reached. See Figure 10 empirically confirming our argument where we train with $\beta$ parameterization instead of $\beta^2$ and initialized with negative values.
>
> Intuitively, $\alpha_h < 0$ term provides information about all tokens except the required matching token. Concurrently, $\beta_h < 0$ ensures that matching any token other than the required one is penalized (i.e., decreases the score). The combination of these two negative influences—the "anti-information" from $\alpha_h$ and the penalizing weight $\beta_h$ applied to the resulting match—yields a **positive composite effect ($\gamma_h > 0$)** that accurately drives the desired matching behavior. This intuitive understanding of **double negation** is empirically verified in Figure 10 in the appendix. Such a double negation is not possible in the task of Nyugen and Reddy, 2025 and therefore there exists a basin $w, \beta < 0$ exists in their framework where ICL does not happen, however it is not the case with us. The analysis is readily applicable to the $u \odot v$ parameterization, which we can use to replace the $\beta^2$ parameterization. The $u \odot v$ form maintains the 2-homogeneity of the original expression and removes the constraint that requires positivity.
>
> **Dynamics with $\beta$ parameterization.** We now investigate the system dynamics when the parameterization in Assumption (A1) is switched from $\beta^2$ to $\beta$. Qualitatively, the core results remain consistent. The conservation law is just modified by constants to: $$\mathrm{d} \left( f(\alpha_1) - \frac{\beta_1^2}{2} \right) = 0.$$However, the change impacts the timescale of emergence. Using the change of variable $s = e^{\alpha} - 1$, the evolution of $s$ under the $\beta$ parameterization is governed by the differential equation:$$\mathrm{d} s = c \sqrt{s^2 + \epsilon^2}$$This yields a timescale of $O(\log{\epsilon^{-1}})$ for $s$ to reach a constant $O(1)$ scale. Consequently, as illustrated in Figure 9, the incremental dynamics are less sharp than in the $\beta^2$ case. Empirically, seeing the incremental dynamics requires initializing the system with a much smaller separation scale. Nonetheless, the incremental dynamics still persist provided the initialization has sufficient separation (see Section F.0.5 for full details).
>
> > **Comparison with Nguyen and Reddy (2025)**
>
> Our initial draft did not elaborate on how our work compares to that of Nguyen and Reddy (2025). Their study investigates the contrast between *in-context learning* (ICL) and *in-weight learning* (IWL) using a simplified task that bears some resemblance to the setting considered in our paper. However, unlike their work, we do not analyze IWL; instead, we focus exclusively on different phases of ICL.
>
> Nguyen and Reddy derive asymptotic dynamics for attention parameters, and these also share certain structural similarities with the corresponding dynamics we obtain. Nevertheless, our analysis differs in two key respects. First, our task necessitates studying a *two-layer Transformer*, whereas their results rely on a *single-layer* model. Second, for the formation of  induction heads in a two-layer transformer, we find that the resulting dynamics deviate substantially from theirs: specifically, our dynamics exhibit *polynomial dependence on the initialization*, in contrast to their *exponential dependence* for ICL.
>
> > **Without all $(k+1)!$ permutations in the context**
>
> To empirically verify the incremental learning dynamics in the case when all the permutations are not present in the context, we train with $k=3$ and seeing only $12$ permutaions (instead of 24) and observe the same dynamics. Please refer to Figure 11 in the appendix.

---

### Author Response · Authors · 2025-12-03
**General Comments**

The reviewers appreciated the phenomenology-driven simplification of the Transformer architecture (R. UrhB) and the subsequent rigorous analysis (R. UrhB, XNc8, g8Xr). The reviewers specifically commended our approach of co-designing the task and the model, noting its elegance (R. XNc8) and methodological soundness (R. jH7v). This unified design successfully yielded a tractable analysis that has potential for actionable practical insights (R. XNc8).

The primary criticism raised by the reviewers concerned the limited experimental validation and the insufficient ablation of the assumptions used in our simplified Transformer architecture. To thoroughly address this key concern, we have added an extensive new section (Appendix F) containing experiments that comprehensively address all points raised.

### **Experimental Evaluation**

We have added the following experiments, which directly address the reviewers' shared concerns.

a) **Empirical validation of theory** — The baseline two-layer transformer model trained under all assumptions (A0–A3). Please refer to Figure 5 and subsection F.0.1 which empirically verifies the results of Theroem 3.1 and 3.2 including the **timescales, directional bias, conservation law**.

b) **Relaxing (A0) and (A3)** — We conduct experiments in which we relax the assumptions of a fixed attention window size (A0) and of restricting attention solely to response tokens (A3). The experiments show the incremental behaviours is retained with relaxing the assumptions (A0) and (A3). See Figure 6 and subsection F.0.2.

c) **Attention only transformer** — We train a attention only transformer with relative positional encoding and show that the incremental learning behaviour is preserved. Furthermore, the trajectory of the transformers second layer parameters, particularly $Q^{\top}K$ looks **very similar to the block structure assumed in assumptions (A1)**. This strongly justifies the assumption. See Figure 7 and subsection F.0.3.

d) **Standard transformer architecture** — We train a standard transformer with value matrices, MLP layers, layer norm, relative positional embeddings. We see the stagewise dynamics in the training and incremental attention patterns in the attention heads of the first layer validating our analysis beyond the simplified architecture. See Figure 8 and subsection F.0.4.

We hope that these detailed ablations, together with the experiments using standard Transformer architectures, will help address the reviewers’ concerns regarding the generality and robustness of our analysis.

**Experimental details.** We have included the experimental details and as a supplementary material we will submit anonymized code for the verification and validity of the experiments.

---

### Meta-Review · Area_Chair_nani · 2026-01-12

**Summary:**

This submission studies the training dynamics of a simplified multi-head transformer architecture for an in-context associative recall task and characterizes the learning order and incremental learning behavior under small initialization. The reviewers raised the following concerns.

* All reviewers commented that the abstracted architecture may be too minimal, and there’s a substantial gap to realistic models. Reviewer g8Xr therefore requested ablation studies for the individual assumptions. The authors partly addressed this concern by including additional experiments in Appendix F.

* Reviewers g8Xr, Xnc8, and UrhB noted the inadequate empirical validation (especially quantitative experiments) of the theoretical findings. The authors conducted new experiments on more standard architectures in the revised manuscript.

* Reviewer UrhB raised the concern that certain assumptions, in particular the squared \beta parameterization, may impact the outcome of the analysis. The authors empirically demonstrated that without 2-homogeneous parameterization, the incremental dynamics still persists (although the transition is less sharp).

Overall, the authors' response addressed some of the concerns raised by the reviewers. However, the area chair believes that this submission would benefit from another round of revision to strengthen the theoretical results and empirical evaluations.

**Reviewer Concerns:**

See above.

**Reviewer Scores:**

Reviewers g8Xr and UrhB both gave a negative evaluation of 2. While their concerns are partially addressed by the new experiments in Appendix F, it is not clear if the current revision is sufficient for a positive score.

---

### Decision · Program_Chairs · 2026-01-26

Reject